# Natural and Cultural Lost Landscape during the Holocene along the Central Tyrrhenian Coast (Italy)

Maurizio D'Orefice [1], Piero Bellotti [2], Tiberio Bellotti [3], Lina Davoli [2] and Letizia Di Bella [4,*]

1 Department for the Geological Survey of Italy, ISPRA—Italian Institute for Environmental Protection and Research, 00144 Rome, Italy; maurizio.dorefice@isprambiente.it

2 Department of Earth Sciences, AIGeo—Italian Association of Physical Geography and Geomorphology, University of Rome, 00185 Rome, Italy; piero.bellotti@gmail.com (P.B.); lina.davoli@uniroma1.it (L.D.)

3 Freelance Archeologist Via Capo Spartivento 13, 00122 Rome, Italy; t_b80@yahoo.it

4 Earth Sciences Department of Sapienza, University of Rome, Piazzale Aldo Moro 5, 00185 Rome, Italy

* Correspondence: letizia.dibella@uniroma1.it

**Abstract:** Landscape evolution over the last 8000 years in three areas located along Tuscany, Latium, and Campania coasts (central Tyrrhenian) has been deduced through a morphological, stratigraphical, and historical approach considering the physical evolution and human activity. Between 8000 and 6000 yr BP, the Sea Level Rise (SLR) dominated and, near the river mouths, inlets occurred. In the Tuscany area, Mt. Argentario was an island and to SE of the Ansedonia promontory a lagoon occurred. The areas were covered by a dense forest and the human influence was negligible. Between 6000 and 4000 yr BP, humans organized settlements and activities, and a general coastline progradation occurred. A tombolo linked Mt. Argentario to the mainland. In the Tiber and Campania areas, coastal lakes and a strand plain developed. Between 4000 and 3000 yr BP, near Mt. Argentario, two tombolos enclosed a wide lagoon. At the SE of the Ansedonia promontory, the lagoon split into smaller water bodies. In the Tiber and Campania areas, delta cusps developed. The anthropogenic presence was widespread and forests decreased. During the last 3000 years, anthropic forcing increased when the Etruscans and Romans changed the territory through towns, salt pans, and ports. After the Roman period, natural forcing returned to dominate until the birth of the Italian State and technological evolution.

**Keywords:** coastal evolution; cultural and land use changes; anthropic impacts; Holocene; Tyrrhenian Sea





## 1. Introduction

The physical coastal landscape is the most sensitive to changes in environmental parameters. It is strongly influenced, also in a relatively short time, by glacioeustasy, particularly in areas prone to load subsidence, tectonics, and/or volcanic activity [1–4]. The landscape changes can be rapid along the coastal plains or deltas. These areas are vulnerable to fast climatic change such as Bond Events or Rapid Climate Change (RCC) [5–10] that are characterized by high variability in frequency and amplitude of storms [11–13] and fluvial solid discharge [14–16]. In the historical period, anthropic forcing was added to the natural one. In fact, human activities have been often settled in coastal areas for the available food sources. Therefore, the landscape was modified over time by the construction of salt pans, fishponds, landings, and ports, and more recently by the intense urbanization often linked to tourist activity [13,17–21].

The central Tyrrhenian coast, between the Argentario promontory and the Garigliano River (Figure 1), shows different morphologies. Rocky promontories (Mt. Argentario, Ansedonia, Capo Linaro, Capo d'Anzio, Mt. Circeo, and Gaeta) and cliffs characterize the coast. The sandy coast is more widespread; it consists of coastal plains characterized by lakes and lagoons (i.e., Orbetello, Burano, Pontini Lakes), deltas (Tiber, Garigliano), and extensive marshy areas, now reclaimed, such as the Pontina Plain. Much of the coast

developed during the last million years on the western edge of a volcanic area (the Vulsino, Cerite, Vicano, Sabatino, Albano, Roccamonfina systems), which today shows only late volcanic signs. In this period, a general uplift affected the area [22]. However, since the Late Pleistocene, the central northern part of this coastal stretch was tectonically stable except for some areas such as the northern sector of the Versilia coast that showed a slight uplift [1,23–26]. The southernmost sector of this coastal stretch (Pontina and Fondi Plains) shows significant subsidence [26], while its southern portion is influenced by the Phlegraean active volcanism.

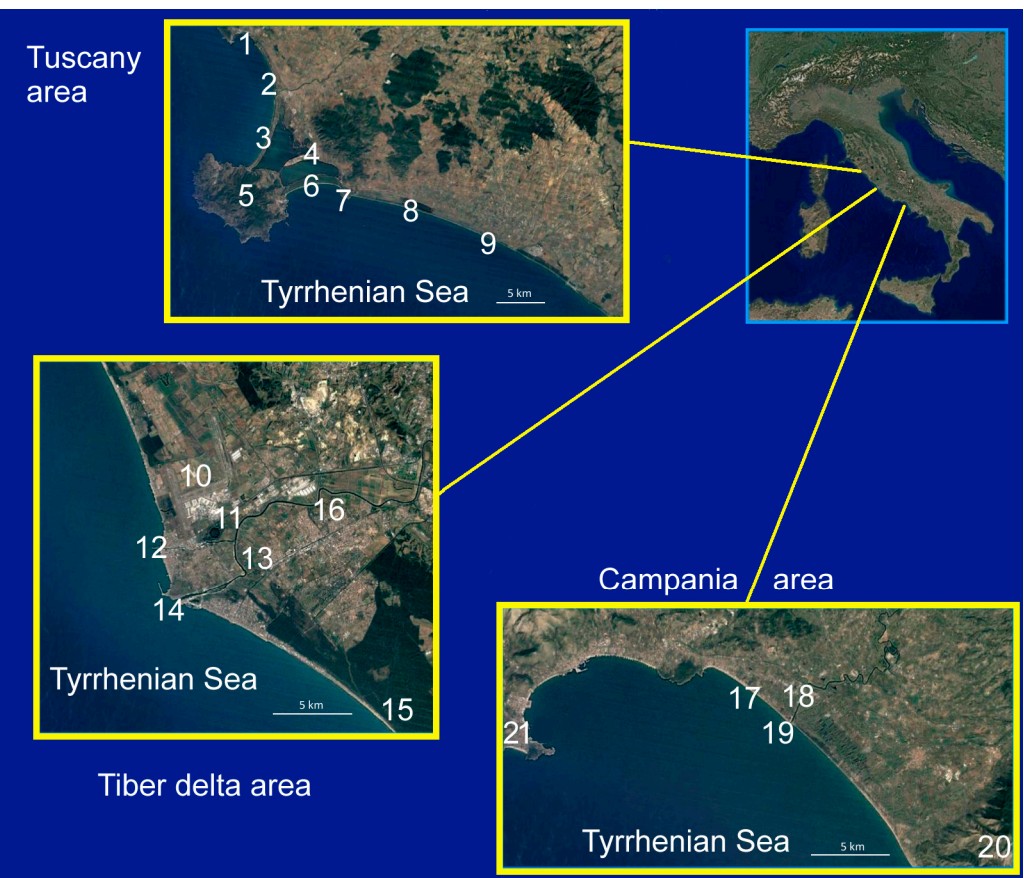

**Figure 1.** Location of the studied areas. *Tuscany area*: (1) Talamone; (2) Albegna River mouth; (3) Giannella Tombolo; (4) Orbetello; (5) Mt. Argentario; (6) Feniglia tombolo; (7) Ansedonia promontory; (8) Burano Lake; (9) Fosso Chiarone mouth. *Tiber delta area*: (10) Fiumicino airport; (11) *Portus* and Trajan Lake; (12) Minor Tiber River mouth; (13) *Ostia*; (14) Main Tiber River mouth; (15) *Laurentum*; (16) *Ficana*. *Campania area*: (17) Mt. d'Argento; (18) *Minturnae*; (19) Garigliano River mouth; (20) Mt. Massico; (21) Gaeta. (Image Landsat/Copernicus-Google Earth).

　　The current morphology is the result of severe changes since the Last Glacial Maximum (LGM). They were due to the SLR, the sub-Milankovian climatic oscillations, the amount of fluvial solid discharge and, partially, local tectonic events [1,2,9,13,14,16,21,22,27]. In addition, during the last 3000 years, anthropic impact became particularly significant. Near the LGM, the shoreline was located about 5–10 km west to its present position. Thus, promontories and some small islands were reliefs on the mainland, which were progressively isolated or submerged for the progressive SLR. At the quasi-still stand, several inlets near each river mouth characterized the coast. In the last 6000 years, the fluvial sediments, more or less abundant according to the sub-Milankovian climatic oscillations, the anthropic activity, and the coastal drift, reconnected islands (i.e., Mt. Argentario), generating lagoons (i.e., Orbetello, Maccarese, Ostia, Minturno), coastal lakes (Pontina lakes), and delta (i.e., Tiber).

During the Holocene, also the activity of humans conditioned the evolution of the landscape. In the central Tyrrhenian coast, although Neanderthal settlements were already known, i.e., Mt. Argentario [28] and Mt. Circeo [29] from the Eneolithic period [30] and mainly from the Bronze Age, the coast is dotted with more or less important settlements [31–33]. Significant coastal settlements developed with the Etruscan civilization north of the Tiber delta (i.e., *Rusel, Tarcuna, Pyrgi*) and the Latin one in the south (i.e., *Lavinium*), which exploited the lagoon areas as a harbor or salt works. The expansion of Rome progressively affected the whole coast with settlements often equipped with ports, increasing the areas devoted to salt extraction. Many coastal settlements were abandoned after the fall of Rome, and marshy areas developed, favoring malaria spreading. This situation persisted during the Renaissance when the coast became dotted with watchtowers. The coast returned to being an important economic and strategic resource only following the reclamation of the area carried out between the end of the 19th century and the beginning of the 20th.

The goal of this paper and the main novelty were to outline the evolution of the landscape over the last 8000 years in three areas (Orbetello-Burano area, Tiber Delta, Garigliano coastal plain; Figure 1) with a holistic perspective that considered, beyond the physical evolution, the presence and anthropic activity from the Bronze Age.

## 2. Materials and Methods

The reconstruction of the landscape is partly based on the revision of data from previous works by the authors, to which reference should be made for the lithological, palynological, faunistical, and chronological details [34–37]. Some data from the literature were also considered (recalled in the discussion of the three areas considered). A new dating (see Table 1) and a new core were added the quotas of all the cores considered were reviewed as well as the environmental interpretation of the different facies. The analysis of the historical evolution of each area described was then added. The paleoenvironmental reconstruction of the three areas was carried out by means of the following approach.

**Table 1.** The values of the $^{14}$C ages used to plot the sea level rise curves of Figures 4, 8 and 11 are reported.

| Tuscany Area | | | | | |
|---|---|---|---|---|---|
| **Laboratory Number** | **Core** | **Altitude** | **Material** | **Age yr BP** | **Age cal. yr BP** |
| Lyon-14233(sacA-49741) | LB3 | −5.00 | Peaty clay | 6335 ± 35 | 7329–7180 |
| Rome-2339 | BU4 | −2.05 | Peat | 3570 ± 40 | 4090–3920 |
| Rome-2342 | BU7 | −1.35 | Peat | 3550 ± 40 | 3900–3720 |
| Rome-2349 | BU10 | −1.30 | Peat | 2790 ± 40 | 2950–2840 |
| Rome-2360 | BU10 | −1.80 | Peat | 4320 ± 40 | 4970–4830 |
| Rome-2364 | BU10 | −2.30 | Peat | 4840 ± 40 | 5620–5480 |
| Rome-2361 | BU10 | −2.80 | Peat | 5310 ± 40 | 6270–5990 |
| Rome-2365 | BU10 | −3.30 | Peat | 5840 ± 40 | 6650–6560 |
| Rome-2350 | BU10 | −3.80 | Peat | 6115 ± 40 | 7160–6820 |
| Rome-2344 | BU12 | −0.50 | Peat | 3700 ± 40 | 4090–3930 |
| Rome-2345 | BU12 | −1.00 | Peat | 4560 ± 45 | 5320–5050 |

**Table 1.** *Cont.*

| Tiber delta area | | | | | |
|---|---|---|---|---|---|
| **Laboratory Number** | **Core** | **Altitude** | **Material** | **Age yr BP** | **Age cal. yr BP** |
| R-1198 α | 150 | −3.24 | Peat | 4710 ± 50 | 5575–5325 |
| R-1198 | 150 | −3.24 | Peat | 4750 ± 60 | 5585–5335 |
| R-887A/α | 150 | −4.24 | Peaty clay | 4640 ± 80 | 5574–5090 |
| R-888 | 150 | −9.44 | Peaty clay | 7730 ± 80 | 8585–8420 |
| R-889 | 150 | −9.59 | Peaty clay | 7770 ± 60 | 8590–8455 |
| R-890 | 150 | −9.74 | Peaty clay | 7930 ± 70 | 8990–8605 |
| LTL-461 a | new | −1.20 | Peat | 1140 ± 40 | 1170–975 |
| Rome-2066 | S3 | −2.93 | Peat | 2720 ± 50 | 2860–2770 |
| Rome-2067 | S3 | −4.03 | Peat | 3465 ± 55 | 3830–3640 |
| Rome-2069 | S5 | −2.95 | Peat | 2555 ± 50 | 2760–2490 |
| Rome-2070 | S5 | −4.15 | Peat | 3375 ± 55 | 3690–3480 |
| Campania area | | | | | |
| **Laboratory Number** | **Core** | **Altitude** | **Material** | **Age yr BP** | **Age cal. yr BP** |
| Rome-2151 | P1 | −2.65 | Peat | 2905 ± 40 | 3170–2990 |
| Rome-2153 | P1 | −3.40 | Peat | 3710 ± 50 | 4150–3930 |
| Rome-2154 | P1 | −3.95 | Peat | 5110 ± 55 | 5920–5730 |
| Rome-2158 | P2 | −2.50 | Peat | 4355 ± 50 | 4980–4850 |
| Rome-2160 | P2 | −4.00 | Peat | 5740 ± 65 | 6640–6450 |
| Rome-2162 | P2 | −5.50 | Peat | 6835 ± 70 | 7730–7580 |
| Rome-2164 | P2 | −7.05 | Peat | 7375 ± 70 | 8330–8040 |
| Rome-2165 | P3 | −1.45 | Peat | 3250 ± 45 | 3560–3390 |
| Rome-2167 | P3 | −4.35 | Peat | 6220 ± 60 | 7250–7020 |
| Rome-2168 | P3 | −5.10 | Peat | 7330 ± 55 | 8180–8030 |
| Rome-2257 | P4 | −2.00 | Peat | 4650 ± 50 | 5440–5050 |
| Rome-2258 | P4 | −2.90 | Peat | 4710 ± 45 | 5580–5200 |
| Rome-2260 | P5 | −1.80 | Peat | 4975 ± 55 | 5850–5610 |
| Rome-2262 | P5 | −3.70 | Peat | 5615 ± 65 | 6450–6300 |
| Rome-2261 | P5 | −5.20 | Peat | 6705 ± 65 | 7660–7300 |
| Not declared | CL1 | −1.50 | Peat | 2503 ± 40 | 2742–2456 |
| Not declared | CL1 | −4.30 | Peat | 6802 ± 45 | 7704–7575 |

Geomorphological analysis. Historical maps available for some areas since the Renaissance were considered including aerial photos starting from the Royal Air Force 1943 surveys, satellite images (Google Earth 2005–2019); LiDAR survey (only for Tiber delta), and field surveys over the last 30 years.

Stratigraphical analysis. Several drillings (by manual, rotary, and percussion mechanical system) and, locally, geophysical surveys were considered. For each sampled sediment, cores were defined as:

- Lithology. Grain-size analysis was performed for the clastic sediments (by sieving and laser diffractometry on fractions > and <62 microns, respectively). Sediments were classified according to [38]. In addition, the main sedimentary structures were considered.
- Faunistic content. Qualitative and quantitative analyses were conducted on samples where the microfaunal content (foraminifera and locally ostracoda) was recorded following the standard procedure.
- Palynology. Although the analyses were carried out by different laboratories for the different areas, they all utilized the same standard procedure for palynological processing.

- Geochronology. The $^{14}$C calibrated ages were calculated using peat and wood samples and, less frequently, shell and bones. Liquid Scintillation Counting (LSC) measurement technique was normally used; but, for materials very low in organic C, Accelerator Mass Spectrometry (AMS) was also used. The conventional ages were calculated according to [39] and reported as yr BP. To take into account the reservoir effect and the past fluctuations of the tropospheric $^{14}CO_2$, the conventional ages were calibrated [40] and given as calendar yr BP time spans. The uncertainty on both conventional and calibrated ages was at the level of $\pm 1\sigma$ (68.2% of probability). Only for the Tuscany area we utilized also the Optical Stimulated Luminescence (OSL) dating of quartz extracts followed standard preparation techniques. Table 1 shows the dates of only the lagoon peaty level used for tracing the SRL curves inserted in Figures 4, 8, and 11.

The historical frame was essentially based on historical sources and on the analysis of data deriving from surveys and archaeological excavations. By comparing the formal and quantitative characteristics of the main productions and settlement traces and correlating the available data, the anthropization patterns resulting from the relationship between the communities and the ecosystem over the centuries under consideration was reconstructed. Table 2 shows the chronology of the historical ages from the Eneolithic to the Renaissance and the life span of the settlements mentioned in the text.



**Table 2.** Chronology of the protohistoric/historical ages and life span of the main settlements mentioned in the text. The Greek Gothic War took place between 535 and 553 AD. The Etruscan civilization started in the ninth century BC.

| Age | Phases | Chronology | Tuscany Area | Tiber Delta Area | | Campania Area | |
|---|---|---|---|---|---|---|---|
| **Eneolithic** | | **5000–3700 BP** | | | | | |
| **Bronze** | **Ancient** | **3700–3500 BP** | Grottino di Ansedonia / Poggio Terrarossa | Le Cerquete-Fianello | | | |
| | **Middle** | **3500–3100 BP** | | | small settlements on dune ridges | | |
| | **Recent—Late** | **3100–3000 BP** | Le Vignole / Punta degli Stretti | Ficana | small settlements in humid costal habitat | | |
| **Iron** | | **2900–2700 BP** | | | Monte d'Argento | | |
| **Archaic** | | **7th–6th BC** | Orbetello | Area rustica Rio Galeria | | Pre Roman Minturnae | |
| **Roman Republic** | | **6th–1st BC** | | | | | Temple of Marica |
| **Roman Empire** | **Early** | **1st BC–1st AD** | Cosa | Portus | Ostia | Roman Minturnae | |
| | **Middle** | **2nd–3rd AD** | | | | | |
| | **Late** | **4th–5th AD** | | | | | |
| **Middle Ages** | **High** | **6th–10th AD** | | Gregoriopoli-Ostia Antica | | Mons Garelianus / Turris Garelianus–Turris ad mare | |
| | **Low** | **11th–15th AD** | | | | Catrum Argenti–Traetto | |
| **Renaissance** | | **16th–17th AD** | | | | | |

## 3. Local Results and Discussion

### 3.1. Tuscany Area (Orbetello-Burano)

Two lagoon systems, (Orbetello Lagoon and Burano paleolagoon separated by Ansedonia promontory) characterize the Tuscan coast between the mouths of the Albegna River and the Fosso Chiarone (Figure 2).

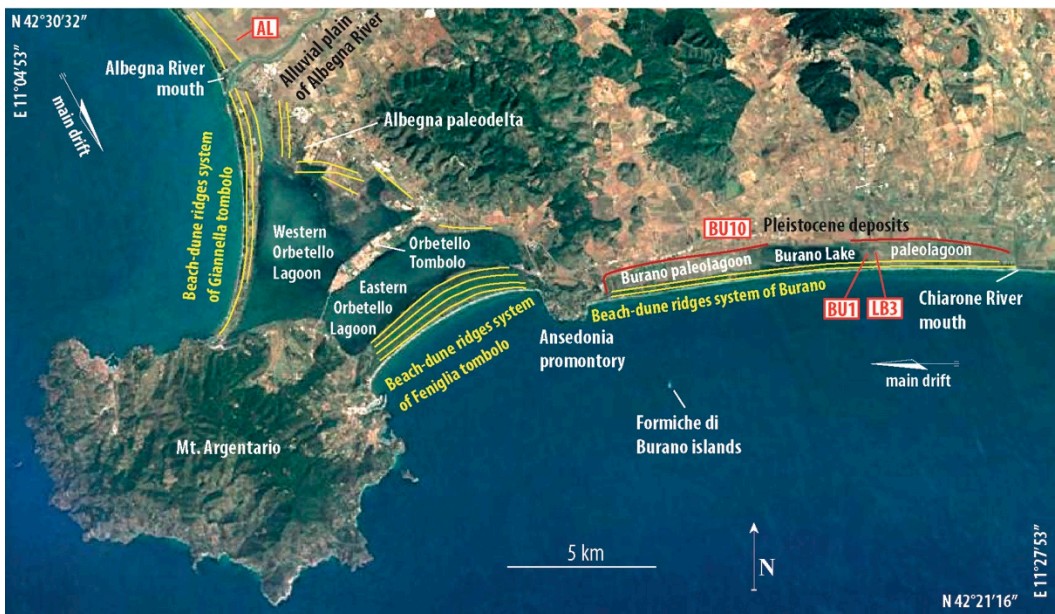

**Figure 2.** Morphological scheme of the Tuscany area. The yellow lines indicate the trend of the Holocene beach/dune ridges. The red line indicates the outline of the Burano paleolagoon. The labels in the white rectangles indicate the location of the boreholes, shown in Figure 3 (Image Landsat/Copernicus-Google Earth).

### 3.1.1. Morphological Setting

The first system (Orbetello Lagoon) is a submerged area, about 27 km² wide with a maximum depth of 2 meters, enclosed between two sandy tombolos that connect Mt. Argentario with the coastal plain between the Albegna river mouth and the Ansedonia promontory.

The northern tombolo (Giannella) is a *spit* 6 km long with a width ranging from about 800 m near the Albegna river mouth and about 300 m near the Argentario where it is interrupted by the only natural inlet of the lagoon (Nassa channel). It is marked by a series of locally discontinuous beach/dune ridges and the inner edge shows washover fans. The southern tombolo (Feniglia), 6 km long, is more regular, has a width close to 1 km, a more organized system of beach/dune ridges dissected by numerous blowouts, and it is rooted landwards at the Ansedonia promontory. Its inner border shows several irregularities related to washover events, which occurred in an initial phase of its formation. Two artificial canals, placed near the roots landwards of the two tombolos, connect the lagoon with the sea and the Albegna River. A third, uncomplete tombolo (Orbetello tombolo), artificially connected to Mt. Argentario, separates the lagoon into two hydraulically connected parts. Several outcrops of rocks and sediments attributable to fluvial, marshy, and paralic *latu sensu* facies characterize the coastal plain. The oldest deposits belong to the upper Pleistocene and are part of the Orbetello Syntema [41]. The Albegna River migrated during the Holocene, leaving meandering paleochannels and alluvial deposits. On the inner edge of the lagoon, a series of beach ridges between the Giannella and Orbetello tombolos mark two little paleodelta cusps connected to the aforementioned paleochannels [41]. Between the Orbetello and Feniglia tombolos only a limited development of the beach ridges occurs. Near the Orbetello tombolo and at the edge of the western lagoon there are limited marshy areas. In the western lagoon, some emerged flat areas and limited sandbanks occur.

The second system consists of a 1-km-wide depression, parallel to the coast with a WNW–ESE trend, limited landward by Pleistocene sandstone deposits and seaward by two parallel Holocene dune ridges. Before the 19th century reclamation, the depression was a wetland periodically submerged into which the Fosso Melone and the Fosso del Chiarone flowed. Currently, the depression is almost flat and at some decimeters below sea level. The depression includes the Burano Lake, the only submerged area after the wetland reclamation, 3.5 km long, 0.5 km wide, with a maximum depth of about 1 m. An inlet, located at the center of the outer side, connects the lake to the sea. Washover fans are particularly evident in the northwestern area of the lake.

### 3.1.2. Vegetation Frame

Between 8000 and 4000 yr BP, the coastal plain landscape was dominated by thermophilous deciduous forest composed of *Quercus cerris*, *Quercus pubescens*, *Quercus suber*, *Carpinus orientalis/Ostrya*, and *Corylus*. *Quercus ilex* is abundant in the surrounding hills followed by Ericaceae, *Olea*, *Phillyrea*, and *Pistacia*. Plants growing under greater control of local edaphic conditions include prevalently herbaceous taxa such as Amaranthaceae and Cyperaceae particularly developed in the perilacustrine areas. A strong increase in herbaceous taxa occurred after 4000 yr BP. Pollen analyses carried out in some lake basins in Tuscany and northern Latium indicated that the variations in vegetation cover in the Neolithic are related more to climatic fluctuations than to anthropic activity [33,42,43]. Only starting from the Bronze Age, the anthropogenic effect on the vegetation appears significant; in the Roman period, the arboreal cover was characterized, similarly to today, by the Mediterranean scrub mixed with oak wood [44].

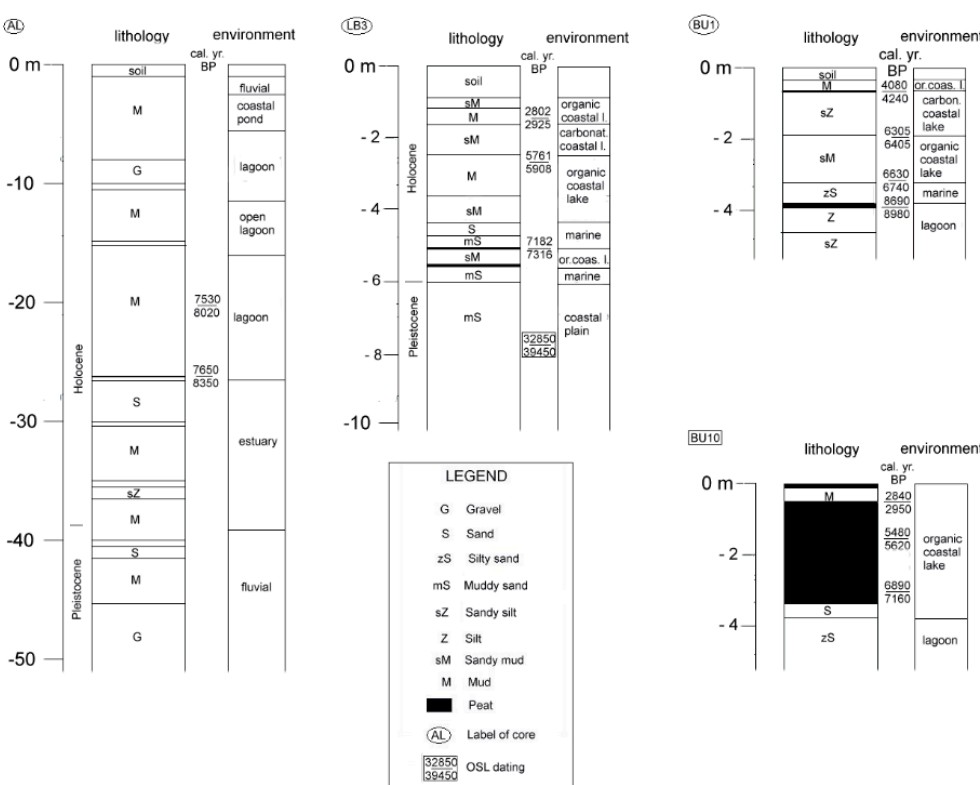

**Figure 3.** Lithological/environmental scheme of boreholes reported in Figure 2 regarding the Tuscany area. AL and LB3 are rotary boreholes; BU1 and BU10 are percussion boreholes. AL is modified after [44], while LB3, BU1, and BU10 are modified after [36].

### 3.1.3. Facies

Six different facies are recognized in this area (Figure 3).

Facies 1. In the subsoil of the Albegna coastal plain, this facies is present between −17 and −10 m. It consists of silty clay and silt with sandy, level intercalations and local thin peat lens. It contains brackish and marine mollusks (*Acanthocardia* spp. and Rissoidae) and meso-polyhaline ostracods (*Cyprideis torosa, Loxochonca elliptica, Cyprideis fischeri, Leptocythere lagunae,* and *Xestoleberis communis*). In the Burano area, the facies occurs at the bottom of the Holocene succession.

Interpretation: Towards the top, the fauna indicates a decrease in salinity (*Cerastoderma lamarcki*), and the presence of sand and fine gravel levels testifies to a greater contribution of fluvial sediment [45]. The facies is attributed to a lagoon/estuary environment.

Facies 2. This facies is well developed in the area of the Burano paleolagoon, which occurs everywhere between the Pleistocene sandstone deposits and the Holocene dune ridges. The facies consists of soft peats, sandy-silty mud, mud with abundant organic matter, and some thin, locally bioturbated silty levels with shell debris. The remains of Posidonia and undecomposed vegetal matter, occasional bivalve fragments, and thin-shell gastropods are present. Ostracods are mainly composed of both brackish (*C. torosa* and *L. elliptica*) and lagoon/coastal (*Xestoleberis dispar*) taxa with variable proportions. They are occasionally associated with a few phytal coastal (*Aurila* spp. and *Leptocythere* spp.) and/or freshwater to low brackish specimens (*Candona angulata* and *Darwinula stevensoni*). A brackish, lagoonal assemblage containing an oligospecific foraminiferal fauna is characterized by the high dominance of *Ammonia tepida* followed by *Haynesina germanica* and *Porosononion granosum* mainly. Locally, in the lower part, more marine taxa (*Nonion* spp., *Triloculina* spp., and *Quinqueloculina* spp.) are present. On the contrary, towards the top, a decrease in foraminiferal content associated with abundant gastropods such as *Hydrobia* spp. and *Planorbis* spp. is recorded. In the lower part, marine dinocysts are also commonly present (including *Spiniferites mirabilis* and *Lingulodinium machaerophorum*), which tend to decrease upward until they are completely absent at the top. Moreover, in the lower part, *Botryococcus* (Chlorophyceae, Chlorococcales/Tetrasporales), a fresh–brackish water colonial green algae, and Terrestrial Fungi are quite abundant. Fungi spores, *Pseudoschizaea, Botryococcus,* and especially *Cosmarium* algae, decrease upward but, unlike the dinocysts, they are still present at the top.

Interpretation: The facies is attributable to an organic coastal lake with a transition from brackish to freshwater conditions.

Facies 3. This facies was intercepted only in the upper part of the cores drilled in the central sector of the Burano paleolagoon. It shows a maximum thickness of 2 m. Whitish $CaCO_3$ (with subordinate gypsum)-enriched silt and scarce fine sand characterize the sediment of this facies only locally interbedded with thin, blackish, peaty levels. Freshwater gastropods (*Hydrobia* spp. and *Planorbis* spp.) and freshwater/low brackish ostracods (*C. torosa, C. angulata, D. stevensoni, Limnocythere inopinata*) are the components of the bioclastic fraction.

Interpretation: The data indicate a carbonatic coastal lake environment where the main components of the NPPs' assemblages are *Cosmarium*, indicative of prevalent freshwater conditions, and *Botryococcus*, which thrives in fresh–brackish waters. Evidence permits us to define a freshwater depositional basin with clear and oligotrophic waters in which peculiar chemical–physical conditions induced an abundant precipitation of $CaCO_3$. The development of this facies started after 6000 yr BP and ended at about 4000 yr BP.

Facies 4. The facies is present between the Albegna River and the Orbetello tombolo both in the subsoil at 3–5-m depth and in the current coastal ponds such as Lo Stagnino. It consists of clayey silt locally with organic matter and a scarce bioclastic fraction. The latter is composed of freshwater and oligohaline ostracofauna (*Cyprideis neglecta* and *Ilyocypris bradyi* with a subordinate presence of *C. torosa* and *Heterocypris salina*) and freshwater mollusks (*Valvata piscinalis, Gyraulus laevis,* and *Bithynia leachi*) [45].

Interpretation: The facies is attributable to coastal pond.

Facies 5. Medium-fine sand with marine taxa (*Cardium* spp., *Glycimeris* spp., *Venus* spp., tellinids, *Quinqueloculina* spp., *Triloculina* spp., *Elphidium* spp., and *Rosalina* spp.)

characterizes this facies. In the northern sector, it forms the Giannella and Feniglia tombolos, whose mineralogical compositions differ in the greater presence of amphiboles in the Giannella site [46].

Interpretation: The facies is attributable to a strand plain and constitutes the beach ridges of some small delta cusps, no longer active, and of the incomplete Orbetello tombolo [41]. Between Ansedonia and the Fosso Chiarone, the facies represents the beach/dune ridges that isolate the Burano depression from the sea.

Facies 6. It is essentially present behind the strand plain between the Albegna River and Orbetello. It shows thicknesses of about 3 m near the river that taper towards the tombolo. It consists of silty clay with an interbedded sandy silt level and lens. Locally calcareous concretions and oxidation tracks are present. Freshwater mollusks (as *Bithynia leachi* and *Lymnaea truncatula*) are rarely recorded.

Interpretation: The described characters permit us to define a fluvial facies.

### 3.1.4. Diachronic Physical Landscape Change

The physical and cultural landscape change is schematized in Figures 4 and 5. Between 7500 and 5000 yr BP, in the northern sector, the SLR interrupted the connection between Mt. Argentario and the mainland [47], leaving only a narrow and incomplete isthmus. The sea, penetrating the Albegna River paleovalley, gave rise to a bay. Around 6000 yr BP, the decrease in the SLR rate allowed for a greater fluvial sedimentary supply to the coast. The bay filled up and the Albegna River began to wander into the coastal plain that was forming. At the end of the period, the river formed a wave-dominated delta cusp north of the incomplete isthmus. To the south of the Ansedonia promontory, a Holocene beach developed in the first phase, separating the Pleistocene coastal deposits from the sea. Further southeast, a narrow lagoon bordered by a thin and discontinuous sandy barrier occurred. The longshore current coming from the southeast carrying the Fiora, Arrone, and Marta rivers' sediments fed the beach and the barrier. As the SLR progressed, the lagoon extended until Ansedonia, the sandy barrier became less thin and discontinuous, and an organic coastal lake sedimentation occurred.

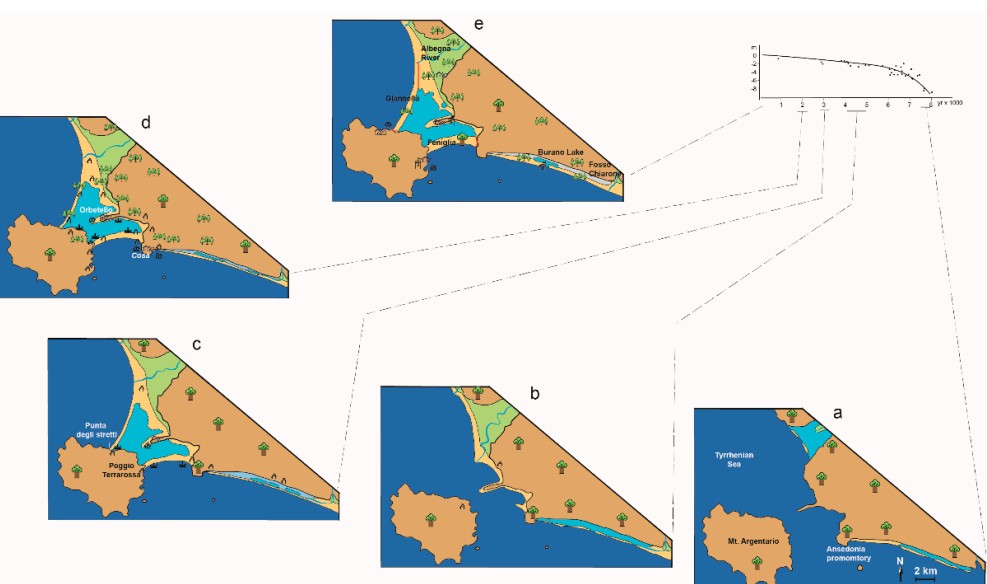

**Figure 4.** (**a–e**). Two-dimensional evolutionary patterns of the Tuscany area inferred from underground and historical/archaeological data. The single images illustrate the coastal landscape relating to the period shown on the abscissae of the sea level rise curve. The curve was plotted based on [14]C dating from Burano paleolagoon peat levels.

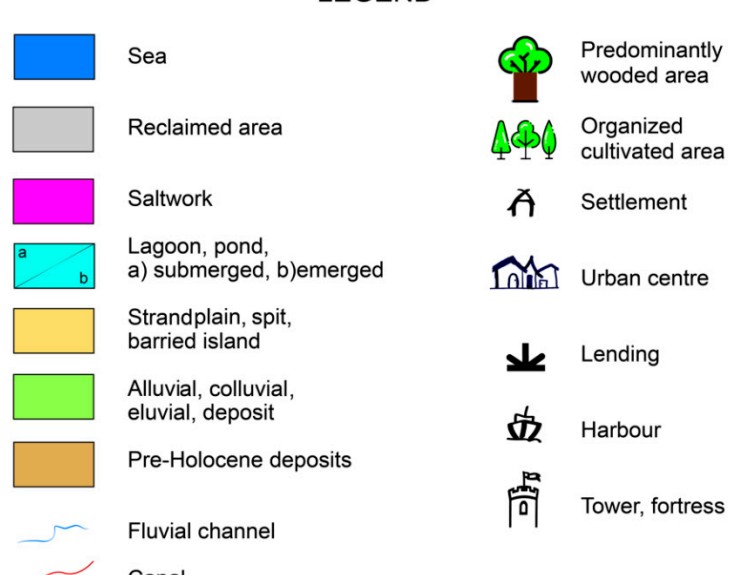

**Figure 5.** Legend for Figures 4, 8, and 11.

Between 5000 and 3500 yr BP, after the quasi-still stand, the sea level rose with minor rates. In the northern sector, the development of the cusp continued and the Feniglia developed completely while the Orbetello tombolo remained incomplete. Southeast of the Ansedonia, the lagoon reached its maximum width (500–700 m) and, along most of its central axis, the sedimentation of a carbonate coastal lake occurred. About 4000 yr BP, the SRL did not change the planimetric features of the lagoon; however, the carbonate sedimentation showed a more discontinuous distribution and progressively disappeared.

Between 3500 and 1000 yr BP, the Feniglia connected Mt. Argentario with the coastal plain, preventing further development of the Orbetello tombolo. The Albegna River migrated northwards on the trace of the current course. It is not clear whether this event was natural or favored by human work [41,48]. The new position of the mouth favored the development of the Giannella that reached Mt. Argentario, giving the Orbetello lagoon its current morphological setting.

In the southern sector, a large part of the lagoon rapidly dried up, or, at least, turned into a wetland locally and periodically submerged so that part of the organic sediments was subject to pedogenesis, forming a thin brown soil. The area of the Burano Lake and a small area close to the Ansedonia promontory remained submerged.

Between 1000 yr BP and the present, south to Ansedonia, there were no substantial changes until the hydraulic reclamation (first part of the 20th century), which definitively dried up the entire area except for the Burano Lake whose depth is currently not over 1 m.

3.1.5. Diachronic Cultural Landscape Change

The morphological and ecosystemic evolutions of the coast between Talamone and Burano Lake marked it up to the entire pre-protohistoric transition. The landscape, today characterized by the presence of the Giannella and Feniglia tombolos, displays a concentration of functional characteristics for anthropization partially comparable to those of the Tiber and Garigliano deltas.

On the basis of the archaeological research, the population appears to be very fluctuating at least up to the end of Ancient Bronze Age, both due to the demographic and environmental evolutions. Some funerary sites, such as the Grottino di Ansedonia [49–51], and some findings of commonly used ceramics, such as Poggio Terrarossa [52,53], are recorded. On the base of limited documentation, the presence of small settlements, scattered around the lagoon, is hypothesized. These settlements were mainly linked to a subsistence based on the consumption of wild plants and shellfish. Therefore, up to this

stage, a low impact on the ecosystem was outlined, due to the presence of small and sparse communities with a predominantly subsistence economy. However, the proliferation and the affirmation of the small perilagunar sites, attributed to the first phase of the Bronze Age, testify to a demographic increase. These sites were probably small, seasonal settlements linked to transhumance [54] and pastoral activities that periodically offered adequate pastures and water.

As in other Italian areas, the Middle Bronze Age for the perilagunar settlements increased and the discovery of new founded settlements represented an important demographic and cultural crossing point. Both the Albegna and Fiora rivers show new, long-lasting settlements located on heights and river terraces, testifying to the first evidence of a territorial system. These will be generalized to characterize the Late Bronze and Iron ages. In this period, the islands of the Tuscan archipelago were also perceived as a territorial segment coherent and functional to the coastal one, beginning maritime activity and routes to Corsica. The exploitation of the islands persisted at least until the first Roman era.

Except for the phase of the Recent Bronze, when a decrease in housing and cultic attestations is evidenced, during the first half of the third millennium BP transition between the Late Bronze and the Iron ages, human presence in the lagoon area was strengthened. A prime example was the settlement of Punta degli Stretti at the junction between the Argentario and Giannella tombolos [51,55–58]. This returned a large amount of evidence of a vital and full-bodied settlement. In this context of cultural change on the Tuscan coast, the first cremation graves characterizing the Villanovan were recorded. The exploitation of the lagoon's food resources and salt and the extraction of metals from the Argentario represented signs of a territorial restructuring persisting until the beginning of the Etruscan era. The Iron Age marked a generalized demographic, cultural, and political change highlighted by the diffusion in most of Italy of Villanovan evidence and a territorial reorganization, which in Etruria was related to the development of *Velch* (Vulci), *Kaisra* (Cerveteri), *Tarcuna* (Tarquinia), and *Vatluna* (Vetulonia) cities. Although the Orbetello coast was also affected by a change in settlements and in the productive–cultural context, it did not suffer a noticeable decline in population. Although the settlement of Punta degli Stretti was ending, the area in question, mainly in the Feniglia tombolo sector [51–53,59–62] of Burano [51], was characterized by various settlements always focused on the exploitation of fish resources, salt, landings, and metals.

From the mid-eighth century BC until the sixth century BC, *Velch*, the hegemonic city of the area, controlled and exploited the territory in a capillary and differentiated way. Moreover, even if some important centers arose, a first network of solitary settlements spread, probably the first agricultural network based on farms.

Among the centers, such as Saturnia or Marsiliana (both along the Albegna River), Orbetello [63] stood out for the exploitation of the natural resources typical of the Argentario area. Orbetello, a fortified town, maintained its importance even beyond the crisis that Etruria went through in the fifth century BC and appeared to be an important reference also after the definitive conquest of the volcanic territory by Rome at the beginning of the third century BC. In the new territorial organization, Rome established *Cosa*, a maritime colony, located near the current Ansedonia, whose *ager* interacted with the Orbetello structures. This condition ended with the Roman conquest of Sardinia that changed the maritime horizons of Rome. *Cosa* played an essential role since the second half of the third century BC: The building of two ports (hydraulically connected by a canal carved into the promontory limestone), south of the Ansedonia promontory and north (*Portus Fenilie*) between the Feniglia and the promontory itself, played a significant role for its logistic and mercantile development. Moreover, its proximity to the Via *Aurelia* [64] and all the local branches made it a reference point for land transport. Alongside the maritime vocation, the agricultural one was increasingly growing. Following the centuriations, a profitable network of *villae rusticae* developed throughout the Republican age, which could exploit both the good coastal soils and the dense and advantageous viability in the mercantile environment. At least until the passage between the Republic and the Empire, the integration between the port structures

was widespread also throughout the Feniglia, dedicated to the export of local wine and oil, and the agricultural area, increasingly organized in rich and well-structured funds. Consistent with this economic framework, it is noted that the population, even up to the beginning of the second century, was distributed in small *vici*, even if *Cosa* maintained an administrative role. From demographic and social points of view, it did not have significant growth [65]. Especially after the Trajan age, the scenario changed significantly, and the Empire incorporated wide and varied territories and markets. The amalgamation of various funds led to *latifundia* formation, characterized by high but scarce productivity. At this time, the activities of the *Portus Fenilie* continued, while the patrician villas destined for *otium* populated the coastal strip. *Cosa* definitely lost its attractiveness, as evidenced by some building collapses [66].

During the Middle Empire, therefore, there was a significant change in the landscape due to new economic choices and more favorable social conditions for a small number of patrician families. In addition, at the same time, the arrival of malaria, as reported by Pliny (Ep. V, 6.2), "*gravis et pestifer ora Tuscorum quae per litus extenditur*", affected the coastal strip. The new agricultural structure with grain estates or pastures was preferred to the previous one, also due to less organization difficulty and less capillary control. All this led to an inefficient management of the waters, which began to invade some areas. The first signs of swamping appeared in this period.

The generalized crisis of the third century AD strengthened the negative economic trend and the demographic decline of the area that was less and less under control. Between the Caracalla and Aureliano periods, it was established *Res Publica Cosanorum* [67], which probably managed the food supply by means of the army.

The fourth and fifth centuries AD marked a definitive decline of the area. The first was characterized by a sharp contraction of the productive and port structures, around which the remaining population gathered, followed by a substantial abandonment. Between the fall of the Western Roman Empire and the Greek-Gothic war, there is no longer any evidence of an institutionalized structure. In fact, only due to the war context of the sixth century AD, the evidence shows an exploitation of some ancient centers for military purposes. The coast at the beginning of the Middle Ages was a sparsely inhabited area, wild and malarial, controlled by the Roman Church, and divided among the main families of princely and bishopric rank. Between the late Middle Ages and the Renaissance, the coast from Talamone to Burano Lake was directly controlled by the Kingdom of Spain, which built a series of fortifications against Ottoman attacks. Subsequently, the war interest for the area decayed, and the zone remained substantially underutilized until the unification of Italy.

### 3.2. Tiber Delta Area

A NE–SW fault system [23,68] characterized the Latium Tyrrhenian margin. This structural configuration determined a lowered and articulated area between Palo and Lavinio, where the Pliocene deposits outcrop. In this area, 100 meters more of Pleistocene sediments and the Holocene Tiber delta were deposited [69]. The delta is the result of a complex evolution starting after the LGM. The reconstruction of the delta evolution was mainly possible by several cores, located along the emerged sector of the delta, and geophysical surveys carried out mainly in its submerged portion [34,69–76]. The Tiber delta shoreline stretches for about 35 km, but the Tiber River sediments affect completely the central coast of Latium.

### 3.2.1. Morphological Setting

The Tiber delta is a cuspate, wave-dominated delta with a strand plain (outer delta plain) and extended land–sea up to 4 km, characterized by beach/dune ridges locally 5 m high (Figure 6). In the strand plain at Capo due Rami, the Tiber River splits into two distributaries. The subordinate one (Fiumicino channel), flowing almost perpendicular to the coast, is the evolution of a Roman canal dredged in the 1st–2nd centuries AD. The main

distributary (Fiumara Grande) at first is parallel to the coast over about 1.5 km, then turns to the WSW with a sharp, elbow-shaped curve and reaches the sea. Until September 1557, Fiumara Grande formed a wide meander directly landward before reverting seaward. Landward, a flat area (inner delta plain), crossed by the Tiber River channel, develops. Here, on the sides of the Tiber there are two ponds now reclaimed.

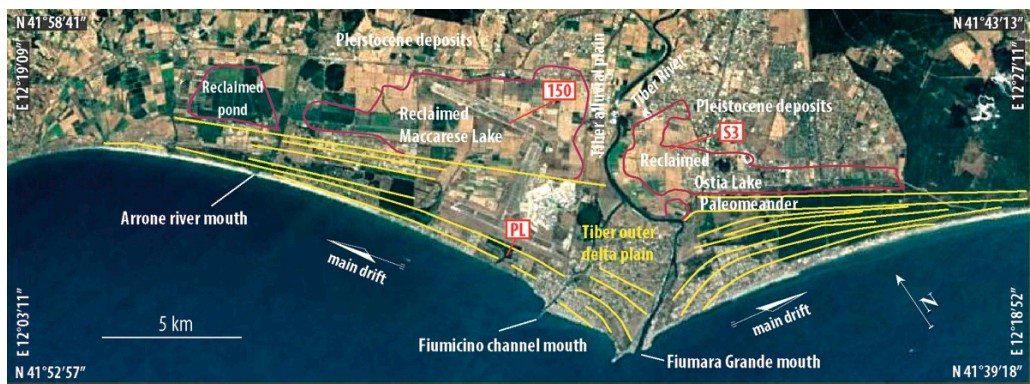

**Figure 6.** Morphological scheme of the Tiber delta area. The yellow lines indicate the trend of the Holocene beach/dune ridges. The red line indicates the outline of the marshes/ponds reclaimed between the 19th and 20th centuries. The labels in the white rectangles indicate the location of the boreholes, shown in Figure 7 (Image Landsat/Copernicus–Google Earth).

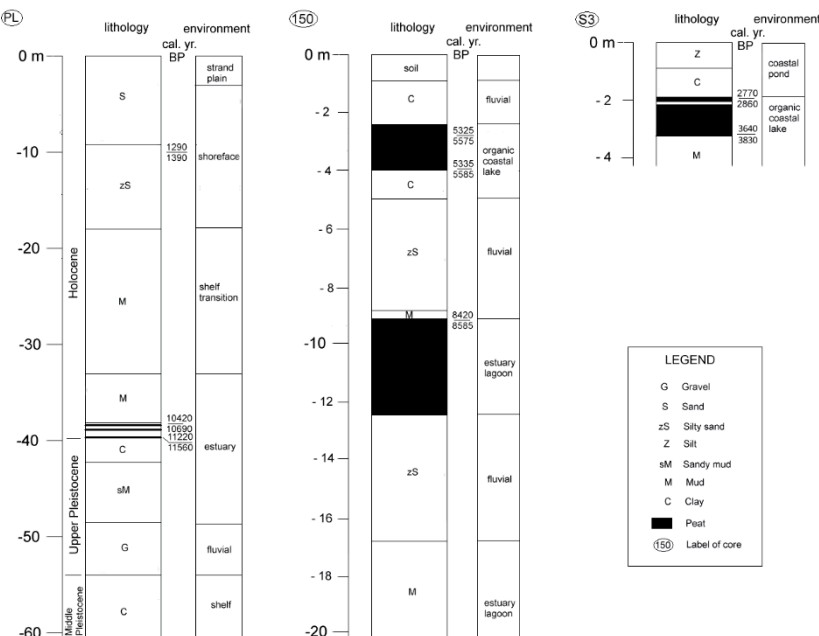

**Figure 7.** Lithological/environmental scheme of some boreholes executed in the Tiber delta area. PL and 150 are rotating boreholes; S3 is percussion boreholes. PL is modified after [68]; S3 is modified after [34].

### 3.2.2. Vegetation Frame

Between about 8300 and 5400 yr BP, the regional vegetation was characterized by dense forests dominated by oaks with a scarce presence of *Olea, Phillyrea, Pistacia, Pinus,* and *Juniperus*, testifying to a limited development of beach/dune ridges. Successively in the northern part, a wide diffusion of riparian trees (in particular, *Alnus* and *Salix*) and a significant presence of *Quercus* evergreen, *Quercus* deciduous, and *Fraxinus* occur up to 2900 yr BP [77]. In the southern part [35], up to 2600 yr BP, a mixed oak-dominated woodland with evergreen elements characterizes the regional context. Around 2400 yr BP, a significant amount of cultivated and anthropochore plants is recorded, including a clear

increase in *Olea*, *Vitis*, and Cannabaceae, along with a progressive appearance of cereals, *Mercurialis*, *Juglans*, and *Castanea*.

3.2.3. Facies

Five facies characterize the architecture of the sediments deposited in the last 8000 years (Figure 7).

Facies 1. Two subfacies constitute facies 1. The first one occurs in the inner delta plain between 5 and 15 m below the ground surface. It constitutes a ribbon-shaped body elongated from land to sea up to the limit with the strand plain. Medium-fine sands, barren in fauna and muddy intercalations with plant remains, mainly characterize it. This lithofacies is embedded in estuary/lagoon deposits (see below). The second unit crops out throughout the inner delta plain, mainly near the modern river channel, with thicknesses between 1 and 5 m. It constitutes a sheet layer of gray-green to brown mud with scarce freshwater fauna, diatomites, and altered volcanic material. In the lower part, there are peaty intercalations with thin shell levels.

Interpretation: The first subfacies is attributed to a bayhead delta body, while the second one is attributed to alluvial plain deposits.

Facies 2. It is present in the subsoil, both in the inner delta plain, below the river sediments, and under the strand plain sediments. It consists of gray-blue mud with thin layers of fine sand often with bioclastic debris or with *Cerastoderma* and *Ostrea* valves interbedded. Plant remains as well as more or less wide peaty lenses are present at different depths. The herbaceous vegetation was mostly composed of sedges and herbs with the presence of *Myriophyllum spicatum*. After 5400 yr BP, Poaceae dominated followed by Cyperaceae and other herbaceous taxa. A peak of micro-charcoal and the presence of cereal-type pollen characterized the northern zone.

Interpretation: The fauna indicates an environment with variable salinity both in time and in space. However, a general trend towards lower salinity is evident both landwards and upwards. The presence of *Myriophyllum spicatum* suggests a large development of areas with fresh waters certainly related to the influence of the Tiber River. After 5400 yr BP, taxa suggest a lowering of the water table probably induced by changes in the sedimentation dynamics of coastal areas [77]. The pollen record from the northern zone provides clear evidence of human impact. These sediments are attributed to deposition in an estuary/lagoon that developed from the beginning of the Holocene up to 5000–6000 yr BP when a further and progressive decrease in marine influence caused a large coastal lake.

Facies 3. In the inner delta plain, above the estuary–lagoon sediments, lies an almost continuous layer of peat, rich in plant remains and barren in fauna. The peat layer has a thickness ranging from 0.5 to over 4 m with the top lying between 2 and 5 m below the ground level and largely settled between 5000 and 2600 yr BP. Pollen data in the northern part [77] indicate, between 5000 and 2900 yr BP, the scarce presence of herbaceous taxa. An important drop of Arboreal Pollen (AP) percentage values, a progressive expansion of Cyperaceae, and an increase in aquatics, mainly *Sparganium/Typha*, *Typha latifolia*, *Alisma*, and Nymphaeaceae, were recorded after 2900 yr BP. A slight increase in Chenopodiaceae around 2600 yr BP occurred. Few are anthropogenic markers indicating cultivation or pastures. In the southern part [35], up to 2600 yr BP, sedge vegetation characterized the environment.

Interpretation: This facies is attributable to an organic coastal lake where the presence of Nymphaeaceae, Lythraceae, *Callitriche,* and *Myriophyllum* highlight a freshwater lake that locally, around 3100 yr BP, showed drying phases (peaks of Asteroideae, Apiaceae, and ferns). Around 2600 yr BP, the definitive disappearance of freshwater hydrophytes, an increase in Chenopodiaceae, and the appearance of *Ruppia* indicate a salinity increase. Anthropogenic markers are negligible.

Facies 4. This facies, 1–3 m thick, developed in the most depressed areas around 2800–2600 yr BP and lies locally on the coastal lake deposits. It consists of sandy, gray to brown, or locally organic muds [78], containing small valves of *Cerastoderma glaucum*, *Hydrobia ventrosa, and Abra segmentum* and some foraminifera (e.g., *Ammonia tepida, Ammonia*

*parkinsoniana, Elphidium crispum*). Locally, there are freshwater mollusks (*Armiger crista, Valva cristata, Lymnea peregra*). Among the pollens, the Chenopodiaceae are abundant and *Ruppia* appears.

Interpretation: The sediments, which were deposited up to the end of the 19th century, are attributed to a sea-connected marsh/coastal pond locally and sporadically were subjected to freshwater inputs.

Facies 5. Present in the outer delta plain, it consists of well-sorted, fine to medium sand, rich in femic minerals. It is about 20 m thick along the shoreline and closes wedge-shaped towards the inner delta plain. In the central area of the delta, it extends for about 5 km in a land–sea direction, decreasing a few hundred meters towards the delta wings. Towards the sea, it lies on neritic sediments and, landwards, on the lagoon/estuary sediments, above described. Shallow, benthic foraminifera (*Ammonia tepida, Elphidium crispum, Lobatula, Ammonia beccarii*) associated with rare reworked planktonic taxa (e.g., *Globigerina falconensis, Globigerina pachyderma, Globigerinoides ruber*) represent the faunal content. Some gastropods, such as *Theba pisana and Colchicella barbara,* appear at the top.

Interpretation: This facies, developing mostly in the last 6000 years, highlights a strand plain with sandbars, beach ridges, and coastal dunes.

### 3.2.4. Diachronic Physical Landscape Change

The different physical and cultural landscapes that have occurred over the last 8000 years (Figure 8) can be outlined as follows.

Between 8000 and 7000 yr BP, during the last phase of the postglacial transgression, a wide estuary/lagoon, partially closed seawards by sandy bars, in which the Tiber, slowly prograding, built a bayhead delta, characterized the area. About 8000 years ago, the sea–lagoon limit ran to the east of Ostia Antica, near Capo due Rami, continuing northwards about halfway through the current Fiumicino airport. The described landscape was maintained at least up to 7000 years ago when the SLR rate rapidly dropped, allowing a rapid Tiber River mouth progradation towards the sandbars.

Between 7000 and 5500 yr BP, a significant landscape change took place. The bayhead delta reached the sandbars, and the waves began to rework the river sediment, making the sandbars more continuous. The lagoon progressively was isolated from the sea, giving rise to two lakes separated by the river course.

Between 5500 and 2900 yr BP, in the first 400 years of this interval, the lake's level decreased. This event could be correlated to a change in the sedimentary dynamics, after the closure of the old lagoon, with consequent overflow of the fluvial sediment in the lakes. In this way, organic sedimentation progressively increased in the lakes, until the end of this interval. An evaporitic level, found in Le Cerquete-Fianello and datable to around 4000 yr BP [79], could be correlated with the 4.2 dry event, which would have temporarily reduced the width of these lakes. The Tiber mouth was probably just north of the current position [35,80,81].

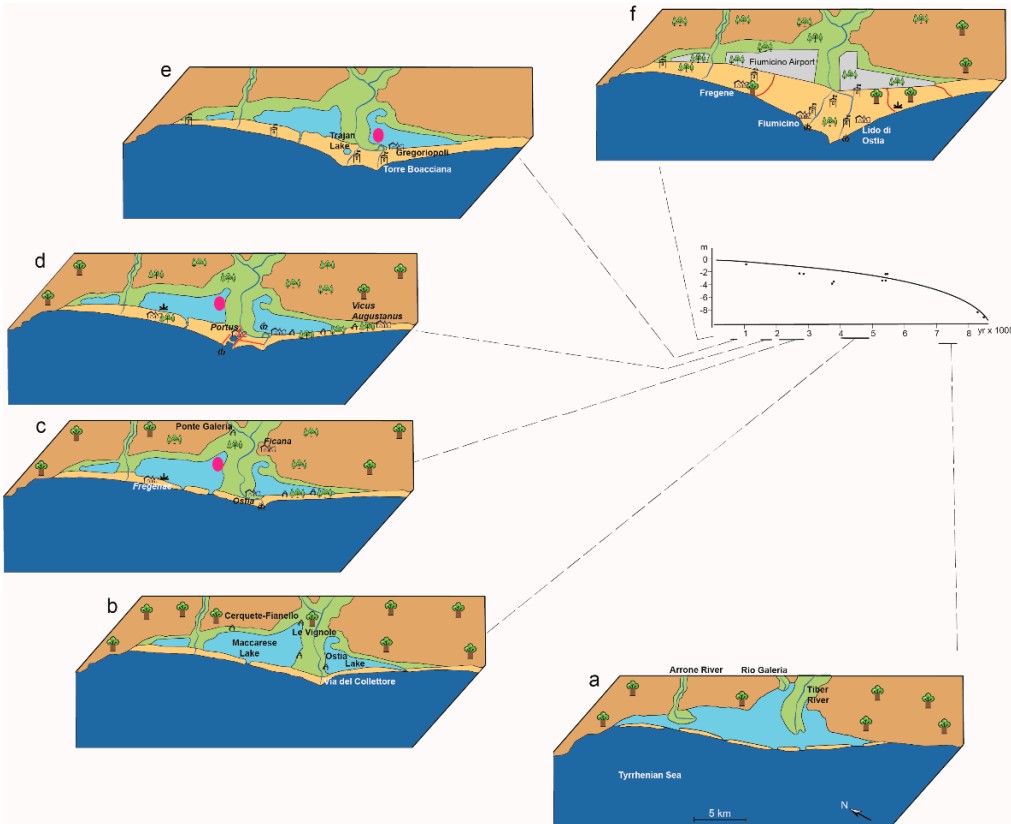

**Figure 8.** (**a**–**f**). Two-dimensional evolutionary patterns of the Tiber delta area inferred from underground and historical/archaeological data. The individual images illustrate the coastal landscape relating to the period shown on the abscissa of the sea level rise curve. The Tiber delta was an area locally subject to repeated changes (excavation of canals and ports and abandonment of meanders) so as to make it difficult to trace a single sea level rise curve [81]. It was, therefore, decided to draw the curve based on only $^{14}$C datings coming from the peat levels of Maccarese Lake. However, those of the Ostia Lake, which are positioned below the traced curve, are also shown in red on the graph.

Between 2900 and 2600 yr BP, a significant change occurred in this period. It is still possible today to observe the river coming from the alluvial valley, instead of heading seawards, and bending landwards, in an anomalous way, before pointing towards the Tyrrhenian Sea with a last elbow. Following this, a meander lapping the southern lake (henceforth Ostia Lake) developed from the Roman period until the 16th century. It is assumed that around 2900–2800 yr BP a break of the left bank, due to a flood or the weak but active local tectonics (the red data of the SLR curves in Figure 8 could indicate a lowering of the left wing of the delta) induced the avulsion southwards of the river. The event caused a local destruction of the beach ridges that isolated the lake from the sea, so that the waters became brackish and silts replaced peat. The abandonment of the previous mouth had to cause a partial erosion of the beach ridges that isolated the northern lake (henceforth Maccarese Lake), which also became brackish with clastic sedimentation.

Between 2600 and 1900 yr BP, in this period the human presence became progressively significant. Starting from 2400 yr BP, the increase in cultivated and anthropochore plants testifies to a progressive increase in the human presence. The new Tiber mouth triggered the progradation of a delta cusp in the position of the current branch of Fiumara Grande. The cusp development was very rapid in the initial phase and, in the fourth century BC, on the left riverbank, between the meander and the mouth, as well there occurred the development of the first Roman settlement in the area (the *Ostia* town). The town developed among the sea, the river, and the Ostia Lake, certainly connected with the sea by means of an inlet located about 3 km south of the Tiber channel. The growth of the cusp was almost



continuous, albeit at different rates, due both to climatic periods and the Roman river basin management [74].

Between 1900 and 1400 yr BP, the territorial setting changed during the Imperial Period mainly due to anthropic activity. In the third century AD, a moderate erosive phase was triggered especially south of the Tiber mouth. This seems to have been caused both by the end of the Roman Warm Period (first BC—second AD centuries) and the cutting of a canal (*Fossa Traiana*), which opened a second and smaller river mouth, moving part of the river sediment further toward the north. However, the erosion did not affect the distal part of the southern wing, where, near *Laurentum*, a limited progradation occurred [82] during the shoreline rectification phases. In addition to the *Fossa Traiana*, several canals were dug to serve a complex port system (Claudius and Trajan harbor, 1st–2nd centuries AD, respectively), which interacted with the longshore current modifying the shoreline. In the first century AD, the evolution of the meander interrupted near *Ostia* the Ostiense road [83].

Between 1400 and 600 yr BP, this period largely coincides with the Middle Ages during which two different climatic periods followed. The cold–humid period (5th–8th centuries; Dark Age Cold Period) with the progressive depopulation occurring after the fall of Rome produced a little progradation and an expansion of the marshy areas in the delta. Subsequently, during the Medieval Warm Period (10th–12th centuries), the shoreline was not subject to significant changes and periodically the minor mouth was obstructed, making the *Fossa Traiana* impracticable. Significant changes also affected the imperial ports. The docks of the Claudius port were partially destroyed, and the basin was partially infilled. Even the hexagonal Trajan basin, no longer connected to the sea, became unusable. The Maccarese and Ostia lakes changed into coastal ponds.

Between 600 yr BP and the present, a new, marked environmental change occurred in coincidence with the cold–humid phase (Little Ice Age), developed above all in the 15th–17th centuries, and ended in the mid-19th century. The major historical floods of the Tiber River were concentrated between 1495 and 1606 AD. The increased energy of the river made both mouths constantly active and conveyed a lot of sediment to the mouths, causing a significant and rapid delta progradation, which assumed a particularly cuspid shape. Several dune ridges developed, facilitating the change of the two coastal lakes into ponds (known as Stagno di Maccarese and Stagno di Ostia). The flood of 1557 AD cut the Ostia meander, which, after 1562 AD, was isolated from the river, assuming the toponym of Fiume Morto. The Claudius port was completely infilled, and the Trajan port became a hexagonal lake. The whole area became malarial until the second half of the 19th century when the reclamation of the two ponds began. The emerged delta reached the maximum expansion at the beginning of the 20th century. In the second half of the century, an erosive phase, largely determined by the construction of hydroelectric basins in the Tiber River catchment, started.

### 3.2.5. Diachronic Cultural Landscape Change

The anthropization of the Tiber delta area was deep and widespread in all the historical phases examined, thanks to both the environmental benefits and the geographical setting. Moreover, the Tiber delta represented the main outlet on the Tyrrhenian coast, as well as a point of contact among Etruria, Apennines, and southern Latium. The Eneolithic and Ancient Bronze ages' human activities were centered on the right bank of the river, whereas the left side, the Ostiense one, was deeply re-elaborated, affecting significantly the state of preservation the archaeological data.

The area surrounding the Tiber River mouth and the Maccarese and Ostia lakes was mainly characterized by fertile land and easy access to fishing areas as well as to springs [84]. In this context, the settlement of Le Cerquete-Fianello [85] developed. The site, located on an offshoot of land that extended towards the Maccarese Lake, gave back important traces of housing structures, hotbeds, material of common use, and a ritual burial of a horse. The stratigraphy of this and nearby sites testifies substantially to a stable presence from the Eneolithic to at least the entire Middle Bronze Age. The territory offered a rich and favorable

ecosystem for both the inhabited area and productive activities, widely exploited during these centuries. The housing model underwent some changes during the Middle and Final Bronze Ages. The site of Le Vignole dates back to this period [86–91] and it testifies to the development of productive activities with seasonal timing. The housing structures were positioned on small artificial mounds composed of sandy fill soil and wooden intertwining that constituted a drainage substrate of the hut walkway. These bumps were periodically renewed, as can be deduced from the succession of sandy layers and the discharge of materials. The anthropic impact had a dual effect due to the optimal access to resources for gathering, fishing, and hunting during the Eneolithic, and the optimal availability of resources suitable for both crafts and sheep farming during the Bronze Age.

The territorial system of the mouth during the whole Middle and Final Bronze Ages, characterized by mostly seasonal and productive settlements, had to be widespread along the Italian peninsula during the whole period. Other environmental conditions displayed other settlement choices. Going up to the hills closest to the coast is where we have evidence of an important settlement, *Ficana*, on the top of Mt. Cugno, a naturally easily defensible area, positioned near a possible ford of the Tiber. *Ficana* [92–96] was a settlement with a stable presence from the end of the Middle Bronze Age up to the Roman conquest in the royal era and which has returned a substantial series of structural evidence, tombs, ceramics, and other objects. Moreover, the delta plain and the surrounding area were characterized by a set of seasonal sites located near the freshwater stretches, on moist and silty–clayey soils. Coinciding with the beginning of the Iron Age, around the 9th– 8th centuries BC, the Maccarese Lake changed into a brackish basin making possible the installation of the Etruscan salt pans, and, at the same time, other important anthropic changes marked the entire area (e.g., the foundation of *Veii*, north to the Rome, in the 10th century BC). The change from a freshwater into brackish basin caused human abandonment along the Maccarese Lake border and the demographic increase along the alluvial fan near the confluence of the Rio Galeria in the Tiber River [97–99]. Here, a small settlement with a rural structure developed. Between the eighth and sixth centuries BC, this characterized the hills near the north bank of the Tiber, between *Veii* and the salt pans, sustaining the economy until the clash with Rome.

The wars, which Rome waged between the eighth and fourth centuries BC, turned out to be a real battle for salt, the main resource of the area. The Roman strategies for territorial control led to the foundation of *Ostia* and the occupation of *Ficana*. According to ancient historiography, both events would be ascribed to Anco Marzio, in the seventh century BC; however, it should be pointed out that the archaeological evidence of a destructive event in *Ficana* dates to the seventh century BC, while the most ancient traces of *Ostia* appear to be those of the castrum related to the fourth century BC.

It is important to note that the area surrounding the Tiber delta began to play a primary role in dominating the salt pans (*Campus Salinarum Romanarum*) and access to the sea, basic elements in the economy of Rome. In fact, starting from the Republican era, *Ostia* played a central role in the logistics and economy of the city, as evidenced by the deep and systematic anthropization of the entire coastal strip that today is included in the Municipality of Fiumicino and Rome (Municipality X). Simultaneously with the enlargement of the urban portion, which already in the Republican era relied on logistics, transport, and commerce, there was a progressive expansion in the southern portion of the countryside, as well as eastern along Via *Ostiense,* which connected *Ostia* to Rome in continuity with the Via *Salaria* [100]. The *ager* that separated the coast from the lake during the first century BC, and a portion of the stretch along the *Ostiense* road, was a very structured rural area with a network of roads and paths and a diversified productive area, including, among others, orchards and farmyard animals (Varrone, De re rustica, Book III) and elements suitable for the drainage of the most humid areas [101]. In addition, in the Republican era context, it is necessary to emphasize that *Ostia* was equipped with landings located along the river stretch, which served the city's *horrea* and served as a hub for Rome.

Moreover, there was a pier located in the wide meander just upstream of today's village of Ostia Antica.

A significant increase in commercial and military traffic characterized the first and second centuries AD. The Tiber delta became one of the main hubs of the Empire and this needed an expansion of the structures toward the north. The port of Claudius was built and later it was expanded and improved with the structure of the Trajan basin. The new harbor center (*Portus*) was surrounded by logistic structures and connected to *Ostia* by canals and roads such as the Via *Flavia*, a northern continuation of the Via *Severiana* [102]. The wide and populous economic pole consisting of *Portus* and *Ostia* changed the area, which was also strongly tested by the clashes of the last phase of the republic age as well as the use of some of its parts. The rustic area, widely structured in the Republican era, was more and more a place of the patrician life. In the strip closest to the coast, some purely peasant spaces were transformed into rich and decorated *villae maritimae*. From the first to the third centuries AD, the territory no longer considered rural was occupied by a necropolis, as well as to the north, today's Isola Sacra.

The generalized crisis of the third century AD caused a contraction of economic activities and a partial abandonment of the areas, evidenced by the substantial closure of the *Ostiense* necropolis. With a progressive economic stagnation during the fourth century AD, the work activities were increasingly limited to the district of the Claudius and Trajan ports, while *Ostia* underwent a further sharp demographic decrease.

The area of the Tiber delta, therefore, was characterized by a slowed economy, with building limited to the reuse of materials and the remodeling of ancient spaces and use starting as a burial area from the fifth century AD (commercial buildings). The few innovations included the building of Christian basilicas either along the peri-urban roads or along Via *Severiana* near the so-called *Villa di Plinio* [103], on a connecting axis with the other streets of the Laurentian quadrant, between fourth and fifth centuries AD.

Germanic raids, the end of the Western Roman Empire, the birth of the Roman-Barbarian kingdoms, and the bloody Greek-Gothic war characterized the troubled period between the fourth and sixth centuries AD. All these factors pushed the inhabitants of the Ostia area to take refuge around the primitive church of S. Aurea. In the ninth century AD, Pope Gregory IV formalized this settlement by defining the new, fortified inhabited area, hence called Gregoriopoli. This new village stood near the last meander of the Tiber, a prominent position on the coast, determining the Saracen attacks in 846 AD. After the success of the Christian league (sea battle of Ostia 849 AD), the Saracens' raids stopped. The new settlement found stability, becoming more and more a reference center both for the management of the salt pans (moved from Maccarese to Ostia Lake) and for the control of river navigation.

Throughout the Middle Ages, the imperial ports became unusable. Archaeological evidence was rather scarce and mainly referred to the remodeling of the fortifications of Gregoriopoli. However, the Tiber delta is reported in different historical sources and archeological proof (i.e., Torre Boacciana) evidenced of stripping of the decorative materials of the ancient city repeated up to at least the 15th century. From the mid-1400s to the mid-1500s, the Ostiense side returned to be the political center. From the archaeological data and, above all, from the study of documents and topographical maps, a first expansion of the village emerged from both the ecclesiastical structures and the defensive ones.

This new impulse also included the reconstruction of the river pier right near the village and the fortress. All these structures represented the papal custom and the trans-shipment point of goods, which, by mean of boats, were sent toward Rome. In 1557, the Tiber changed course by cutting the meander, which gradually became silted up and swamped; the pier became unusable, and customs was moved first to Torre Boacciana and, successively, to Torre San Michele.

The meander, later called the dead river, and the ancient coastal lakes became malarial areas, and the residual population was directly linked to the salt pans; the reversal trend occurred only after the unification of Italy through the reclamation and elimination of salt pans.

### 3.3. Campania Area (the Garigliano Coastal Plain)

The Garigliano delta plain developed, between Latium and Campania, within a wide Quaternary extensional basin belt that stretched from the Tyrrhenian margin to the Central–Southern Apennine chain [3,104,105]. The coastal plain was characterized by the terminal stretch of the Garigliano River that built its delta on Plio-Pleistocene sediments and tephra (from Campanian and Roccamonfina volcanic centers). These deposits infilled a graben developed during a time interval ranging from the Miocene to about 125,000 yr BP [106–109].

#### 3.3.1. Morphological Setting

The Garigliano River separates two strand plains, trending NW–SE, characterized by wet and depressed areas (Figure 9). The inner strand plain is referred to as the Eutyrrhenian [110], and the outer one is part of the present (Holocene) Garigliano River delta. The northern triangular depression is locally up to 1.5 m b.s.l. Its shorter side parallels the river through about 800 m, and, in turn, the longer one stretches about 1.4 km along the Holocene strand plain. The trapezoid shape of the southern depression is up to 1.2 km wide, extends about 4 km parallel to the coast, and it is locally deepening to 2.5 m b.s.l. Historical maps show that up to the 18th century both depressions were partially submerged, whereas today, after the land reclamation, they are totally dry. Based on aerial photographs, paleochannel traces, recording the past wandering of the Garigliano River, were recognized in the northern depression between the present river channel and the southern depression and along the eastern side of the Eutyrrhenian strand plain. However, the numerous Roman witnesses along the river prove that the river course has not changed at least since Roman Times. The Holocene strand plain shows beach/dune ridges 2–3 m high to the north of the Garigliano River and up to 9 m high to the south.

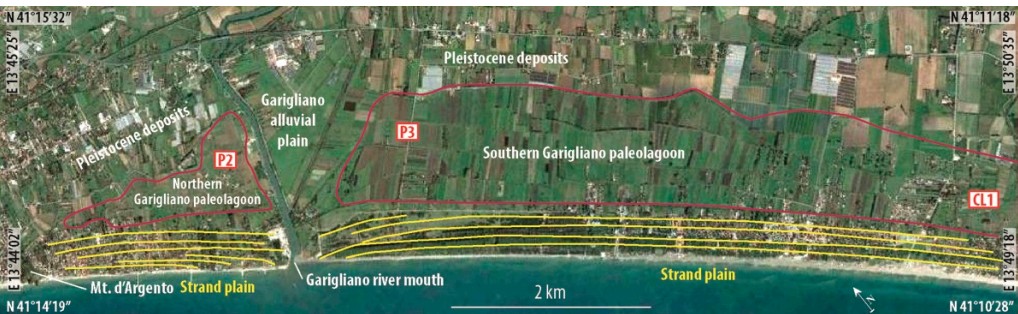

**Figure 9.** Morphological scheme of the Campania area. The yellow lines indicate the trend of the Holocene beach/dune ridges. The red line indicates the outline of the marshes/ponds, today reclaimed. The labels in the white rectangles indicate the location of the boreholes, shown in Figure 10 (Image Landsat/Copernicus—Google Earth).

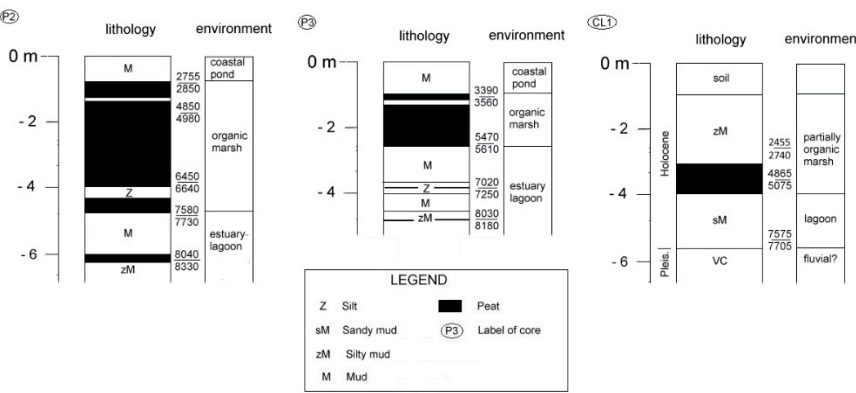

**Figure 10.** Lithological/environmental diagram of some boreholes carried out in the Campania area. P2 and P3 are percussion boreholes; CL1 is rotating boreholes. P2 and P3 are modified after [36]; CL1 is modified after [111].

### 3.3.2. Vegetation Frame

The appearance of vegetation and its evolution in the Garigliano area are discussed in detail in [36,111,112]. Between 8200 and 5800 yr BP, in a coastal plain, a large forest cover, mainly consisting of *Quercus* deciduous, *Alnus*, *Salix*, *Populus*, and *Juniperus*, occurred. The presence, in the first phase, of abundant *Alnus* closest to the current river mouth suggests a riparian forest. Subsequently, oak woods expanded, and the Mediterranean taxa increased. Towards the end of this period in the innermost areas, a mosaic of freshwater and brackish water environments coexisted with open environments and mixed woods located far from the site. Between 5800 and 3100 yr BP, forest cover was more present with considerable incidence of hygrophilous trees (*Alnus* and *Salix*), which also extended into the southernmost portion of the coastal plain, and an increasing of the Mediterranean taxa such as *Olea and Castanea* and coprophilous fungal spores occurred. Between 3100 and 2700 yr BP, the forest cover decreased with minor incidence of *Salix and Juniperus* types and the Mediterranean taxa. In contrast, *Populus* as well as several freshwater aquatics' plants spread. The *Olea, Juglans,* and *Castanea* (OJC group) included all the three trees, and cereals occurred.

During the Roman and post-Roman time, *Alnus* and *Myrtus* increased, while *Salix* and *Quercus* deciduous decreased. In the OJC group, *Juglans* was absent and *Castanea* increased. In the modern age, forest cover dropped notably, owing to the decline of riparian trees and the Mediterranean taxa. Broadleaved trees remained steady and the complete OJC group was maximized.

### 3.3.3. Facies

Five different facies developed in the last 8000 years (Figure 10).

Facies 1 developed between the end of the postglacial sea level rise and the early stage of the subsequent quasi-still stand. Near the Garigliano River, it consists of dark-gray mud levels with brackish and marine fauna (*Ecrobia ventrosa, Cerastoderma glaucum, Abra segmentum, Mytilus galloprovincialis, Parvicardium exiguum, A. tepida, Haynesina germanica, H. depressula,* and *Porosononion granosum)*, wood debris, peat, and gray silt intercalations. Farther from the river channel are present mud and silt with local peaty levels or mollusk fragments and sandy bed with parallel lamination with *Bittium reticulatum, Loripes lucinalis, Abra segmentum, and C. glaucum,* in addition to *A. beccarii, A. tepida, A. parkinsoniana, L. lobatula, H. germanica,* and Cyperaceae, Poaceae, and Salicornia pollens. Dinoflagellates are in the southernmost part of the coastal plain [112], where this facies lies on the "Campanian Ignimbrite" volcanic deposits [111].

Interpretation: The facies highlights a lagoon/estuary rich in hygrophilous woody vegetation with probable fluvial inputs. In the southern part, fauna and pollens reflect an ongoing lagoon infilling under variable marine and fluvial inputs.

Facies 2. It consists of a homogeneous peat suite, devoid of fauna but with well-preserved vegetal remnants (*Alnus, Salix*) that witnessed the spreading trend in hygrophilous woods. Furthermore, gray-greenish silts, sometimes peaty and locally with pumice grains, and gray medium-fine sand with parallel laminations occur.

Interpretation: The sediments reveal an organic coastal lake in which clastic levels locally break the organic sedimentation and, according to the fauna remnants, witness washover or crevasse events. This facies largely developed between 5500 and 3500 yr BP.

Facies 3. It is characterized by dark-gray or brown mud with local peat levels and rare intercalations of gray-ochre silt levels with occasional laminations and pumice grains. The fauna only includes rare freshwater gastropods such as *Acroloxus lacustris*, *Bithynia tentaculata*, *Gyraulus laevis*, *Planorbis planorbis,* and *Valvata cristata.* The pollen content mainly consists of Callitriche, *Nymphaea alba* type, *Potamogeton,* and *Myriophyllum.*

Interpretation: The facies is attributable to a freshwater basin (marsh/coastal pond) governed by decantation and occasional flooding inputs. It is believed that drying phases alternated with shallow, limpid ephemeral ponds prior to the final drying up. The development of this facies began around 4000 yr BP and tended to shrink after the Roman period until it disappeared following the land reclamation in the 19th century.

Facies 4. It spread mainly near the present Garigliano River. It includes gray-greenish mud levels with thin, reddish silt laminae. Locally, calcrete nodules, small and rounded pebbles, pumice, subfossil tree roots, and rare brick fragments occur.

Interpretation: This is a fluvial facies.

Facies 5. The sediments are characterized by coarse sand with polygenic pebbles and marine bivalves changing upwards in medium-fine sand containing subrounded or bladed carbonate and volcanic clasts. Sand is interbedded with a decimetric, dark mud level with altered plant remains. Locally, marine shells and low-angle parallel lamination occur. At the top, there is well-sorted, fine-medium ochre sand locally exhibiting cross lamination.

Interpretation: The sediments mark the Holocene strand plain.

### 3.3.4. Diachronic Physical Landscape Change

The stratigraphical data allow us to reconstruct the evolution of the local physical and cultural landscape for the past 8000 years (Figure 11).

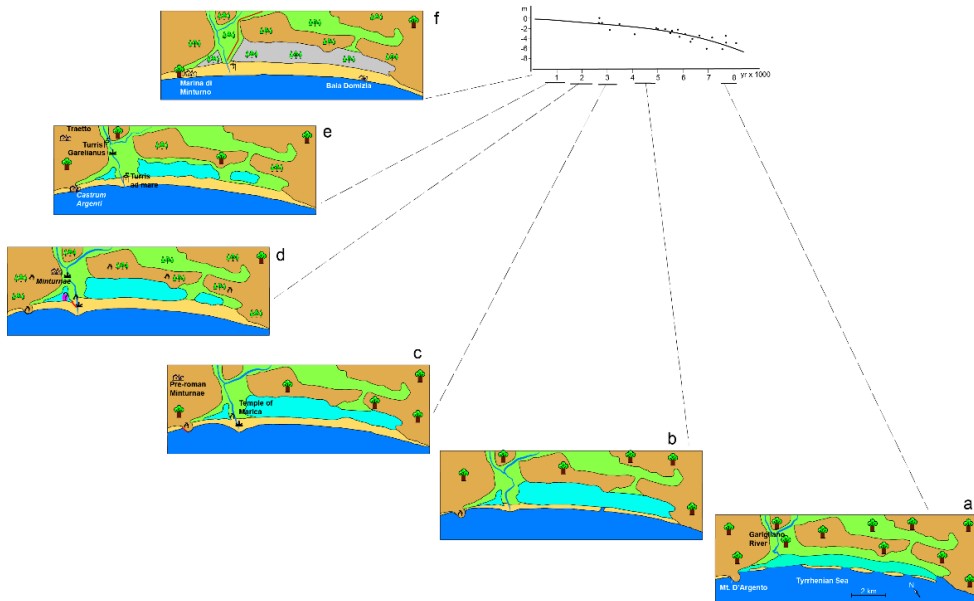

**Figure 11.** (**a**–**f**). Two-dimensional evolutionary schemes of the Campania area inferred from underground and historical/archaeological data. The individual images illustrate the coastal landscape relating to the period reported on the abscissa of the sea level rise curve drawn mainly based on [14]C dating of marsh/pond peat levels [36].

Between 8000 and 7000 yr BP, a partially emerged wet zone, with a riparian wood, was present in the northern zone. On the contrary, proceeding south, a sheltered, narrow lagoon, locally well connected with the sea, developed. Only landward, a continental environment influenced by alluvial deposition occurred.

Between 7000 and 5500 yr BP, during the final transgression stage and the beginning of the sea level still stand, the lagoon migrated landward. In the northern zone, a coastal barrier bordered a freshwater coastal lake with organic sedimentation. In the southern zone, the lagoon, less and less connected with the sea for the expansion of the coastal bars, progressively extended, enriching itself in organic sediment and producing a local mosaic of fresh and brackish water environments.

Between 5500 and 2600 yr BP, at the beginning, a delta cusp developed in the northern zone, turning the coastal bars into a strand plain throughout the area, and a freshwater coastal lake developed in the southern part. Brackish environments progressively disappeared, and the river wandered between the twin lakes. Perhaps due to the end of the 4.2 yr BP dry event, a greater input of alluvial sediments progressively changed the coastal lakes in coastal ponds/marshes, with prevailing clastic sedimentation. Woodlands declined, and the pollen records reveal an early anthropic impact.

Between 2600 yr BP and the present, two ponds/marshes separated by the river course and by a partially eroded delta cusp characterized the early landscape. Some hundred meters wide, a strand plain bordered the ponds. Human impact on the territory increased but the pollen and fauna data do not show the use of the ponds as salt works. The significant presence of *Myriophyllum alterniflorum* suggests that clear and oligotrophic water filled the ponds, probably because of the proper water management in Roman times. The subsequent lack of maintenance of the drainage system in the Middle Age resulted in enhanced solid inputs to the ponds, with consequential phases of partial drying. The ponds were never filled in, and the areas now must be kept dry by ongoing reclamation. Forest cover decreased contemporary to the reduction of the wetlands and the expansion of agrarian systems. Similarly to other deltas of the Tyrrhenian coast [113,114] from the Roman period, progradational and erosive phases followed one another, and currently the delta cusp is missing.

3.3.5. Diachronic Cultural Landscape Change

Over the millennia, the area of the plain of the Garigliano delta has maintained a set of natural characteristics suitable for human settlement and different types of exploitation.

Throughout the period from the Neolithic to the Middle Bronze Ages, this area had scarce signs of population but was set in an interesting context, as exemplified by the Neolithic sites in the Mt. Massico area, attributable to the Laterza culture [115], and by the later ceramic material discovered on the marine side of the same mountain.

During the Middle Bronze Age, the inhabited areas on the fertile plain disappeared, whereas those on the slopes were scarce. In some centuries (first half of Middle Bronze), the small settlements located preferentially on the dune ridges or on land with pozzolana, always near water pools, became more numerous. A flora useful for feeding, craft, construction, and potentially phytotherapeutic fields (plants with tannins and salicin are included) characterized this habitat. The availability of terrestrial and lake fauna must also be considered, as well as the possibility of exploiting for housing purposes the highly draining soils, sand and pozzolana, and, for the clay production, the more humid areas with clayey silts. At the end of Middle Bronze Age there was a gradual increase in settlements on elevated zones' ground, among the first were the areas surrounding Mt. Massico. In this period and in the following centuries, the exploitation of the area changed in a stable and widespread manner. In addition, the increasing population led to a cultural and social impulse determining a political, economic, and territorial control of the area. The data for the Recent and Late Bronze Ages show an increase in small sites located in a humid coastal habitat. The accumulation of commonly used materials in small stratifications and without particular traces of structures leads us to consider these sites as productive with periodic

attendance. The proto-urban centers exploited the different environmental and geomorphological characteristics of the entire area. The socio-political balance in the Late Bronze Age involved the appearance of small agricultural and pastoral settlements on fertile land without natural defenses. As in other contexts (for example, Punta degli Stretti, see Tuscany area), parallel to the above-described structure, autonomous centers located along the coast arose. This was the case of the town on the top of Mt. d'Argento [31,116,117], which was built close to the sea and, at the same time, in an elevated position. The coastal stretch had an increasingly important role, as a point of contact with the Mycenaean world. Many demographic variations were associated with the constant exploitation of the area, similarly to Mt. Massico, a site that characterized the transition to the first Iron Age. Between the ninth and seventh centuries BC, there was also important cultural evidence [117]. Starting from the eight century BC, the archaeological data show varied typology of settlements, which, nevertheless, seemed to remain in the wake of the territorial management described above. From the seventh century BC, well-defended centers developed on topographically high areas, becoming a reference for the scattered settlements in the productive plain. In the sixth century BC, attendance, already attested to on Mt. d'Argento, was stable and large. The presence of areas fortified by polygonal walls suggests a capillary military control by means of communication (routes and passes). These *oppida*, extending along the entire stretch of the Garigliano River, seem to have had control and defense roles of the population [118]. This function, known since the sixth century BC, was strengthened up to the end of the third century BC, when the area was involved in the Samnite wars. The historical sources report that the population, the Aurunci, settled in the three main *urbes*: *Minturnae*, *Vescia* (south of Mt. Massico), and *Ausona* (near today's Ausonia). These *urbes* were probably the afferent centers of *oppida*, *vici*, and farms, as indicated by Tito Livio. In continuity with previous centuries, the coastal area played an important sacral role, as testified to by the first monumental temple dedicated to the goddess Marica [119–126] that was an important sacral structure for all Mediterranean populations [120]. This temple was close to the dune ridges and oriented towards the lake.

From the sixth century BC, significant changes occurred in Italy. The Aurunca population was exposed to the growing Roman power, which determined the conquest and Romanization of the Garigliano coastal plain by the third century BC. After the suppression of the revolt at 314 BC and the destruction of *Ausona, Vescia*, and *Minturnae*, the territorial structure was redesigned on the basis of a new foundation of the city, roads (i.e., *Appia* road, a consular road that partially followed pre-existing paths near the coast), and centuriation.

In 296 BC, the reorganization process was completed by the establishment of two further colonies of a *castrum* type, defined *coloniae maritimae* (Lev. XXVII, 38, 3-5), under Roman law: *Minturnae* and *Sinuessa* (near today's Sessa Aurunca). *Minturnae* was located on the Pleistocene dune on the right bank of the Garigliano River about 2 kilometers from the coastline, and *Sinuessa*, near the coast south of Mt. Massico. Their locations show the clear intent to occupy spaces to acquire the maximum control, and, so, to guarantee stability to the Romanization policies. This led to a widespread population and exploitation of rural potential, as indicated by *Corpus Agrimensorum Romanorum* of Igino (Hyg. De Lim. Const., 178, 6–9) and confirmed by the analysis of aerial photos. Throughout the middle and late republics, for the proliferation of rustic and patrician villas along the coast and the hinterland, the area began a vital and populous territory. The urban areas started an evident monumentalization while the rural one was terraced to facilitate both building and farming. The coast of *Minturnae* became the reference point for all the main agricultural production of the Garigliano basin. Particularly the wine production assumed international significance, as testified to by the *Minturnae* remains along the routes to Gaul, Spain, and the Middle East [127]. The recovery of various logistics and shipbuilding structures and a pier suggests that the fulcrum of this flourishing activity was the harbor. This scenario grew and settled, down to the whole of the first century AD, including even the perilagunar area, as evidenced by the presence of many *villae maritimae*, remains of structures related to fishing, fish farming, and *garum* production. In addition, epigraphs for the *salinatores*

members certified the salt production is in the urban area [128]. Similarly to other equally prosperous *agri*, the territory gradually was structured in wide estates managed by the exponents of the main Roman families. The turbulent centuries of the second half of the republic saw *Minturnae* at the center of many events, among which were the servile revolt and the flight of Gaius Marius. Successively, the colony and its territory suffered considerable and repeated damage. Between the first BC and first AD centuries, there was a substantial reorganization, as reported by Igino in the *De Limitibus Constituendis* (Hyg. De Lim. Const., 177, 8–15; 178, 1–9). In the first century AD, a new territorial parceling in the south of the river ensured new prosperity, so much so that many of the *villae* turned into real patrician residences. Until the whole second century AD, *Minturnae* was undergoing a major overhaul. In addition to the improvement of the recreational, cultural, and worship structures, there was also a modernization of the harbor and the mouth. In fact, probably logistical structures from the first century AD were identified. The strong mercantile characterization of the area led us to suppose the existence of a harbor also in the lagoon area [129–134], although there is no archaeological evidence. The second century AD, as in most of Italy, marked a moment of less imperial control, and a progressive impoverishment of economic and building activities ensued.

Additionally attributable to this phase was the new reconstruction of the ancient temple of Marica [119], now oriented towards the river and, therefore, disconnected from the pond. The center of commerce and wealth that moved all this building fervor, and perhaps the arrival of new cults, such as that of Isis [119], had to play a more heartfelt role than the tradition.

In the third century AD, the crisis of the Empire reached its peak and the economy underwent a general contraction; consequently, *Minturnae* suffered a significant impoverishment. The sepulchral reuse of the commercial area was an eloquent testimony of the situation until the fourth and fifth centuries AD. The demographic collapse and the Greek-Gothic war marked the end of *Minturnae*, which, in a few decades, lost all economic or political importance. The last traces are those relating to the lime production centers through the recycling of building material. In 590 AD, a letter from Pope Gregory the Great attested to the passage of the town church to the diocese of Formia, an evident sign of depopulation. Italy, since the sixth century AD, was under Lombard dominion, with the exception almost exclusively of the territory governed by the pope. The Garigliano River was its border [135]. The alluvial plain was depopulated; the few inhabitants looked for safe areas suitable for a subsistence economy. The Minturnese *ager* was dotted with small settlements, often grown around the ancient *villae rusticae* or positioned on the plateaus already exploited in protohistoric times. Until the 10th century, the only vital centers were Gaeta and Sessa Aurunca.

A discontinuity with respect to the set of small towns under the direct influence of the papacy was the settlement witnessed by some historiographical, toponymical, and partly archaeological traces of a Saracen group. In the last two decades of the ninth century AD, historical sources mention *Mons Garelianus* [136] as the town exploited for the control of the coast and piracy carried out by the Saracens. The ancient port structures were exploited, suitable for the seafaring activity that scourged the entire Tyrrhenian coast. The grip of Christian cities allied under the aegis of the Pope, which since the mid-ninth century AD clashed with the Saracen forces, suffocated the base of the Garigliano that was abandoned following the battle of 915 AD [137]. It followed that the coast acquired an important military control role; the first watchtowers were built in the aftermath of the Christian victory at Garigliano. *Turris Gariliani* [137,138], on the right bank of the river, already erected at the end of the ninth century AD reusing Roman structures, was repaired following the battle, and *Turris ad mare* [138] was erected on the left bank between 961 and 981 AD. With a progressively more stable political situation, the towers took on a role of control of the mouth of the river, which, being still navigable, connected the coast with the inland centers; this made them a point of interest for part of the population (i.e., Mt. d'Argento, *Castrum Argenti*) [139,140]. Some fortified structures agglomerated inhabitants around them. From

the 12th century AD, for about two centuries, there was a wall, a church, and some houses, as well as a road connecting the fortified plateau and the town of *Traetto*.

These settlements remained isolated throughout the Middle Ages, and the water management was very difficult under the Little Ice Age meteorological worsening. The progressive swamping pushed the resident communities to abandonment. Until the unity of Italy, the area was uninhabited, acquiring agricultural and tourist interests only after the reclamation of the coastal ponds.

## 4. General Discussion

The evolution of the coastal landscape of the three studied areas shows both common elements and significant diversities. The latter are due to both the peculiar geological and morphological characteristics of each area (presence of islands, headlands, river mouths) and the different anthropogenic activities (socio-political and commercial), which the areas acquired during the historical times.

In the period between 8000 and 6000 yr BP, the SRL rate, although decreased with respect to the previous period, limited the sedimentary supply to the coastal belt. A dense forest dominated by deciduous oaks with a local presence of riparian tree covered the three areas (Figures 4a, 8a and 11a). The population was scarce and the human influence on the environment was negligible. Near the river mouths (Albegna, Tiber, Garigliano), the coastline inflected, generating more or less wide inlets/estuary that were slowly being filled. In the Tuscany area, Mt. Argentario was an island close to the coast, and to the SE of the Ansedonia promontory the shoreline migrated landwards and formed a long and narrow lagoon locally well connected with the sea (Figure 4a). In the Tiber area, the wide estuary, bordered by barrier islands partially migrating to the land, contained the bayhead Tiber delta (Figure 8a). In the Campania area, to the north of the Garigliano, a small freshwater lake was forming; in the southern part, a barrier, partially discontinuous and slowly migrating to the land, bordered a lagoon locally connected to the sea (Figure 11a).

Between 6000 and 4000 years ago, the SLR rate, stabilized at values close to 1 mm/yr, was no longer a limit to the sedimentary supply of the coastal area. The river sediments, reworked by the longshore currents, produced a general progradation of the coastlines that caused changes of the landscape in each three areas. In the Tuscany area, the pollen content indicates that the variations in vegetation cover in the Neolithic are related more to climatic fluctuations than to anthropic activity [33,42,43]. The Feniglia developed and in the SE of the Ansedonia promontory, and the long lagoon become wider (Figure 4b). A carbonate sedimentation, probably linked to the rise of hydrothermal fluids, partly characterized the well protected from the sea lagoon [37]. The territory, still covered by a dense forest with the development of herbaceous plants in the perilagoon areas, was inhabited by small and sparse communities with a mainly subsistence economy, which settled, scattered close to the lagoon areas; in this regard, a comparison was proposed [33] with the site of Le Cerquete-Fianello of Maccarese. In the Tiber area, the bayhead delta reached the barrier islands, causing their coalescence and the transformation of the wide inlet into two coastal lakes (Figure 8b). The river separated the lakes, characterized by fresh water, mainly towards the inner margin.

The progradation of the Tiber mouth gave rise to a delta cusp. The area was vegetated by a dense forest of oaks and riparian trees and subordinately by a dune vegetation. The evolution of the physical landscape brought an environment favorable to human stable settlements. The development of the Eneolithic Le Cerquete-Fianello site on the northern lake is evidence of that for its long stability and for the activities that well exceeded the subsistence economy. In the area of Campania, the sediments of Garigliano allowed the development of an almost continuous barrier that separated a long coastal lake from the sea.

The river wandered between the lakes and progressively produced a weakly cuspate strand plain (Figure 11b). In the oak forest, the hygrophilous plants and Mediterranean taxa increased. The small population produced only minimal settlements in the plain.

Between 4000 and 3000 years ago, the physical evolution of the territory was determined by the sedimentary contribution to the coast. Although some areas changed significantly, a more widespread anthropic presence occurred. In the Tuscany area, the dune systems developed, and Mt. Argentario was permanently connected to the coastal plain by two tombolos that enclosed a wide lagoon. At the SE of the Ansedonia promontory, the long coastal lake progressively filled up, splitting into smaller water bodies (Figure 4c).

The vegetation was enriched with herbaceous plants and dune vegetation. The demographic increase was evidenced by some stable settlements developed on the fluvial terraces and in proximity of the coast for the control of the new marine routes. Considering the Punta degli Stretti site, which displayed settlement similarities with the Puntata di Fonteblanda site near Talamone, we can hypothesize a well-defined management method, in which these centers controlled the main resources and the maritime communication routes. Moreover, small, mostly seasonal, perilagoon settlements proliferated, exploiting the water and meadows' supply. The physical landscape of the Tiber area had no great variations other than an extension of the delta cusp and a limited reduction of the coastal lakes. The vegetation was enriched with riparian trees, especially in the northern part, and dune vegetation. For some centuries, the Le Cerquete-Fianello site was active but, towards the middle of the period, the use of the territory was characterized by small, seasonal peri-lacustrine settlements, not exclusive of the northern lake, which devolved essentially to farming and handicrafts (Figure 8b). At the same time, in a dominant position on the hill between the Tiber and a southern lake, a new stable site began to develop. In the Campania area, the variation of the physical landscape consisted of an increase in the strand plain and a partial burial of the coastal lakes (Figure 11c). Forest cover and water plants decreased while some cereals extended. Mostly seasonal settlements developed in the plain close to the lakes and the river that represented productive sites suitable for grazing [141]. In the second half of the period, stable settlements rose on the elevated areas at the edge of the plain even though they continued to control and use the seasonal productive sites. The proto-urban root of Lazio can be identified in the agglomeration of the population in well-defined and well-defended areas, testifying to a persisting territorial attraction from productive and economic points of view.

The coastal landscape showed both significant physical and anthropic evolutions starting from 3000 yr BP. Social and political changes had the first important effects on human–environmental interaction. In the Tuscany area (Figure 4c,d), the physical landscape began as very similar to the current one: Mt. Argentario, well connected to the coastal plain by the two tombolos and the Albegna River mouth, permanently had a position at the eastern root of the Giannella. At the SE of the Ansedonia promontory, the ancient coastal lake evolved in a wet zone well isolated from the sea by a continuous dune ridge. Starting from the Bronze Age, the anthropogenic effect on the vegetation appears significant, and, in the Roman period, the arboreal cover was characterized, similarly to today, by the Mediterranean scrub mixed with oak wood [44]. The demographic increase involved the growth of small settlements located mainly both around the Burano Lake, for the production of salt, and Feniglia, now dotted with landing places. In this period, Etruscans organized the territory. On the central tombolo, Orbetello, the first fortified urban center, developed. In its surroundings, an organized agricultural system and the use of the lagoon were undertaken. In the second part of the millennium, the Roman control extended on the territory. On the Ansedonia promontory, *Cosa* town was founded. The city provided landings along the entire Feniglia by means of ports located on both sides of the promontory that were connected by a canal carved into the mountain. Centuriation, road networks, and agriculture, characterized by well-structured funds, developed. In the Tiber area (Figure 8c), the river moved the mouth further south where a new delta cusp developed, while the previous one was partially eroded and the two lakes became brackish. At first, the Latin center of *Ficana* and the northern one by the Etruscan center of *Veii*, which implanted the salt pans in the Maccarese Lake, controlled the southern part of the delta. With the expansion of Rome, the salt pans remained, *Veii* and *Ficana* disappeared, and *Ostia* with its

river port was born near the Tiber mouth. A road network developed as well as a series of *villae* for agricultural production and breeding. In the Campania area (Figure 11c,d), there were no significant variations in the physical landscape. The northern lake took on an important cultic function and some fortified centers were built in the area. Towards the end of the period, the Roman conquest modified the landscape through the centuriation, road development, and the foundation of *Minturnae*. This urban center provided a river port from which the agricultural products departed.

In the first millennium of the Common Era, two distinct phases of coastal landscape evolution followed one another. In the first half of the period, anthropic action was an important forcing, but it was significantly reduced in the next phase. In the Tuscany area (Figure 4d), at first the coast became home to patrician villas and the agricultural structure evolved towards *latifundium*. However, this pattern did not favor territorial management, and, in the central centuries of the period, a phase of swamping began, causing a decrease in the population and port activity. For the rest of the time the area was marshy, malarial, and sparsely populated. The Tiber area was deeply affected by human impact in the early part of the period (Figure 8d). On the northern part, salt pans were active and ports were built. All these activities involved commercial structures and roads, as well as the building of canals that determined a double-mouth delta, partially modifying the coastal dynamics.

On the southern part, the agricultural organization displayed large villas that followed one another along the coastal road. These villas represented both productive and leisure centers for the rich owners. After the third century, the population slowly decreased, shrinking strongly at the end of the empire. In the early Middle Ages, the great salt pans were no longer productive and the ports were partly buried. The territory, not carefully managed, became marshy, and a small town, close to oldest *Ostia*, gathered a small number of inhabitants. In the Campania area (Figure 11d,e), in the first period, *Pagus* and *vici* created the political pattern to manage the first agricultural production structures, already conceivable as real farms. Commercial, maritime, agricultural, and salt production continued. It is believed that the salt pans were located at the mouth of the Garigliano [142], but in a freshwater context of problematic localization. The trace, detectable in an aerial photo of a narrow channel extending from the northern lake towards the river, as well as for the medieval salt pans of *Ostia*, could suggest the presence of a small salt pan. As with the other areas, the coast hosted the patrician villas that replaced the rustic villas. However, after the third century, as the result of a demographic decline and the fall of the Roman Empire, *Minturnae* disappeared. In the early Middle Ages, a subsistence economy was established again and the few inhabitants settled in small, elevated centers abandoning almost completely the plain, subject to swamping and river floods.

In the first part of the last millennium (late Middle Ages), the natural change in physical landscape was limited and human influence poorly affected the coastal landscape. In the second part of the last millennium, some areas showed a remarkable change in the physical landscape due to a significant human–environment interference occurring mainly in the last two centuries. In the Tuscany area, slight variations were related at the edge of the Orbetello lagoon. The sparsely populated area was dotted with towers and coastal defenses in the middle of the millennium. Only in the last century (Figure 4e) was the Burano area reclaimed, determining an agricultural increase, whereas, in Orbetello, ichthyoculture and tourism were on the rise. The history of the Tiber area was different (Figure 8e,f). Here, already in the first centuries, salt production began and continued until the 18th century, no longer in the Maccarese Lake but in the Ostia Lake. The imperial ports disappeared, but, as with the lake, the Trajan dock remained. Starting from the 15th century, the rapid progradation of the Tiber mouths, associated with a final modification of the main channel, expanded the strand plain that was enriched with towers guarding the mouths. The scarce population used the territory mainly for pasture. At the end of the 19th century, reclamation drained the ponds, the population increased, and agriculture extended; in the 20th century, tourism and urbanization strongly developed. The area that represented, during the Roman period, the connection between Rome and the world through the port structures returned to

have the same function in the 20th century by the Fiumicino airport. In the Campania area (Figure 11e,f), the situation occurring *quo ante* persisted until the middle of the millennium, until, because of the worsening climate, a phase of progradation occurred.

This fact produced an extension of the strand plain and a further hydraulic disorder. The whole area was almost abandoned and used for grazing. At the beginning of the 20th century, the reclamation of the coastal ponds [143] and the Garigliano damming allowed agricultural use and a limited tourism development.

## 5. Conclusions

In the final analysis, different forcings caused the evolution of the coastal landscape in the considered areas over the last 8000 years. They effected differently in time and space. The natural forcing was closely linked to the Milankovian and sub-Milankovian climatic variations to which, during certain periods, anthropic activity overlapped. If the SLR dominated in the first two millennia, the amount of sediment, coming to the coast, dominated over the next two millennia when, locally, humans began to organize settlements and activities using the resources of the territory. Human impact progressively increased, mainly in the Tuscany and partially in the Tiber areas, when the social and political structure of the Etruscans deeply modified the territory mainly through the salt pans and the organization of the ports. During the Roman period, the anthropic influence increased, highlighting firmly the relationship between landscape change and socio-political organization of the population that inhabited it. The landscape of the Tiber area, close to the center of power, resulted as the most impacted. The degradation of that organization caused the renewed dominance of natural forcing. Only the restoration of the socio-political organization of the new Italian State, together with the technological evolution of the last two centuries, has allowed limiting the effect of natural forcing.

**Author Contributions:** Conceptualization, M.D., P.B., L.D. and L.D.B.; Methodology, P.B., T.B. and L.D.; Supervision, M.D., P.B., T.B., L.D. and L.D.B.; Writing—original draft, M.D., P.B., T.B., L.D. and L.D.B. All the authors participated in writing, reviewing, and editing the manuscript. All authors have read and agreed to the published version of the manuscript.

**Funding:** This research received no external funding.

**Institutional Review Board Statement:** Not applicable.

**Informed Consent Statement:** Not applicable.

**Acknowledgments:** We would like to thank Silvana Falcetti and Laura Di Pietro for the drawing of figures.

**Conflicts of Interest:** The authors declare no conflict of interest.

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
