# Peer review of "Natural and Cultural Lost Landscape during the Holocene along the Central Tyrrhenian Coast (Italy)"

_land, doi:10.3390/land11030344_

Round 1

Reviewer 1 Report

The authors have amended their original manuscript considering my comments and other reviewers'. Overall, the manuscript has improved and is eligible to be published on Land.

I don't feel qualified to judge on the English language, but I would suggest a last revision. Some new pieces of text may be non-standard or lack of style (e.g. line 660, 921).

Author Response

Thank you for the suggestions. We followed them:

We checked and improved the english Language modifying the sentences related to lines 660 and 921. 

Reviewer 2 Report

Dear Editor,

The paper “Natural and cultural lost landscape during the Holocene along the Central Tyrrhenian coast (Italy)” by Maurizio D’Orefice, Piero Bellotti, Tiberio Bellotti, Lina Davoli and Letizia Di Bella has been modified, accomplishing the Reviewers' suggestions. The paper is improved and suitable for publication.

Regards

Luigi Bruno

Author Response

Thank you for the comments

Reviewer 3 Report

Thank you very much for good work and interesting article.

Author Response

Thank fo the comments